# Multiple hominin dispersals into Southwest Asia over the past 400,000 years

Huw S. Groucutt[1,2,3 ✉], Tom S. White[4], Eleanor M. L. Scerri[5,6,3], Eric Andrieux[7,8], Richard Clark-Wilson[8,9], Paul S. Breeze[10], Simon J. Armitage[8,11], Mathew Stewart[1], Nick Drake[10,2], Julien Louys[12,13], Gilbert J. Price[14], Mathieu Duval[12,15], Ash Parton[16,17], Ian Candy[8], W. Christopher Carleton[1], Ceri Shipton[18,19], Richard P. Jennings[20], Muhammad Zahir[2,21], James Blinkhorn[5,8], Simon Blockley[8], Abdulaziz Al-Omari[22], Abdullah M. Alsharekh[23] & Michael D. Petraglia[2,12,24,25 ✉]

Pleistocene hominin dispersals out of, and back into, Africa necessarily involved traversing the diverse and often challenging environments of Southwest Asia[1–4]. Archaeological and palaeontological records from the Levantine woodland zone document major biological and cultural shifts, such as alternating occupations by *Homo sapiens* and Neanderthals. However, Late Quaternary cultural, biological and environmental records from the vast arid zone that constitutes most of Southwest Asia remain scarce, limiting regional-scale insights into changes in hominin demography and behaviour[1,2,5]. Here we report a series of dated palaeolake sequences, associated with stone tool assemblages and vertebrate fossils, from the Khall Amayshan 4 and Jubbah basins in the Nefud Desert. These findings, including the oldest dated hominin occupations in Arabia, reveal at least five hominin expansions into the Arabian interior, coinciding with brief 'green' windows of reduced aridity approximately 400, 300, 200, 130–75 and 55 thousand years ago. Each occupation phase is characterized by a distinct form of material culture, indicating colonization by diverse hominin groups, and a lack of long-term Southwest Asian population continuity. Within a general pattern of African and Eurasian hominin groups being separated by Pleistocene Saharo-Arabian aridity, our findings reveal the tempo and character of climatically modulated windows for dispersal and admixture.

As the only land bridge between Africa and Eurasia, Southwest Asia occupies a unique position for understanding key stages of human evolution and the peopling of the planet. Changing environmental and ecological conditions at the shifting interface between the Saharo-Arabian and Palaearctic biomes strongly influenced patterns of human demography through the isolation, diversification and subsequent mixing of populations[4,6–11] (Supplementary Information, section 1). A prominent example concerns the geographical context of Neanderthal–*sapiens* admixture. Although it has been suggested that this occurred in Southwest Asia owing to the ubiquity of Neanderthal ancestry in humans outside Africa[6], 'on-the-ground' evidence for admixture, or even spatial and temporal contemporaneity with *H.*

*sapiens*, has remained elusive in the region. One reason for this is the severely fragmented nature of Southwest Asian palaeontological, palaeoenvironmental and archaeological records. This has in turn limited our ability to overcome problematic generalizations regarding the palaeoanthropological record of Southwest Asia and address key questions about the extent to which hominin occupations of the region were continuous, the role of hominin dispersals into and within the region, and how these dispersals and interactions between hominin populations related to changes in biogeography, environment and ecology.

Research in Southwest Asia has traditionally focussed on deeply stratified cave sequences in the Levantine winter-rainfall woodland zone[11–15] (Fig. 1, Supplementary Information, section 1). This has led to

[1]Extreme Events Research Group, Max Planck Institutes for Chemical Ecology, the Science of Human History, and Biogeochemistry, Jena, Germany. [2]Department of Archaeology, Max Planck Institute for the Science of Human History, Jena, Germany. [3]Institute of Prehistoric Archaeology, University of Cologne, Cologne, Germany. [4]Department of Life Sciences, Natural History Museum, London, UK. [5]Pan-African Evolution Research Group, Max Planck Institute for the Science of Human History, Jena, Germany. [6]Department of Classics and Archaeology, University of Malta, Msida, Malta. [7]Department of Archaeology, Durham University, Durham, UK. [8]Centre for Quaternary Research, Department of Geography, Royal Holloway University of London, Egham, UK. [9]Department of Geography and Environmental Science, University of Reading, Reading, UK. [10]Department of Geography, King's College London, London, UK. [11]SFF Centre for Early Sapiens Behaviour (SapienCE), University of Bergen, Bergen, Norway. [12]Australian Research Centre for Human Evolution, Griffith University, Brisbane, Queensland, Australia. [13]College of Asia and the Pacific, The Australian National University, Canberra, Australia Capital Territory, Australia. [14]School of Earth and Environmental Sciences, University of Queensland, Brisbane, Australia Capital Territory, Australia. [15]Geochronology and Geology, Centro Nacional de Investigación sobre la Evolución Humana (CENIEH), Paseo de Atapuerca, Burgos, Spain. [16]Human Origins and Palaeoenvironments Research Group, School of Social Sciences, Oxford Brookes University, Oxford, UK. [17]Mansfield College, University of Oxford, Oxford, UK. [18]Institute of Archaeology, University College London, London, UK. [19]Centre of Excellence for Australian Biodiversity and Heritage, Australian National University, Canberra, Australia Capital Territory, Australia. [20]School of Biological and Environmental Sciences, Liverpool John Moores University, Liverpool, UK. [21]Department of Archaeology, Hazara University, Mansehra, Pakistan. [22]Heritage Commission, Ministry of Culture, Riyadh, Saudi Arabia. [23]Department of Archaeology, College of Tourism and Archaeology, King Saud University, Riyadh, Saudi Arabia. [24]Human Origins Program, National Museum of Natural History, Smithsonian Institution, Washington, USA. [25]School of Social Science, University of Queensland, St Lucia, Queensland, Australia. ✉e-mail: hgroucutt@ice.mpg.de; petraglia@shh.mpg.de

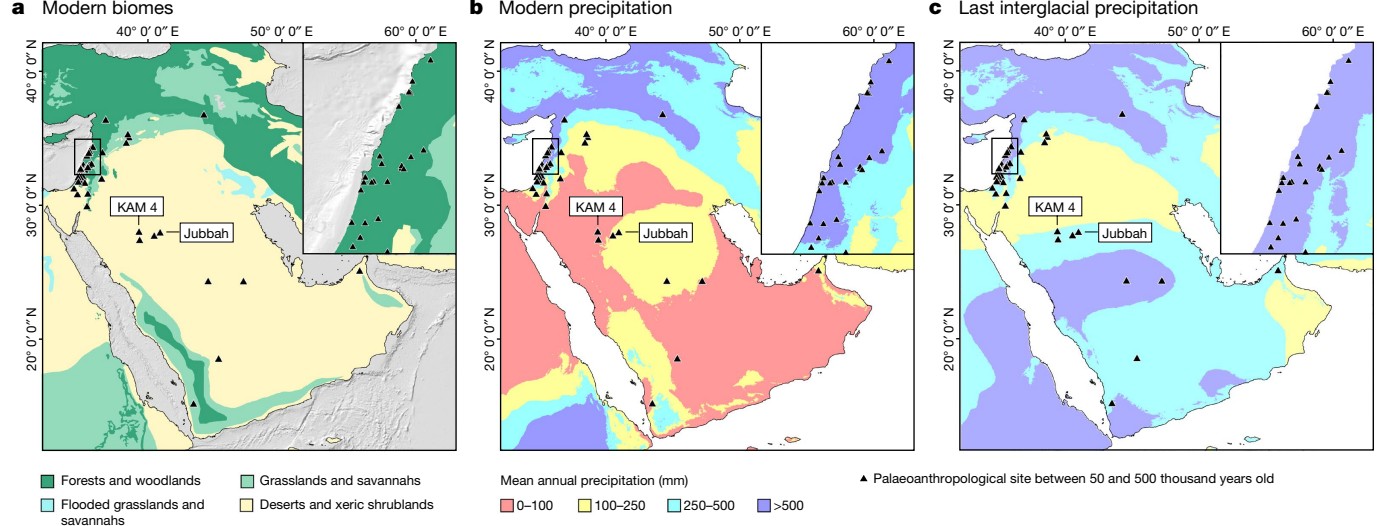

**a** Modern biomes **b** Modern precipitation **c** Last interglacial precipitation

Forests and woodlands Grasslands and savannahs

Flooded grasslands and savannahs Deserts and xeric shrublands

Mean annual precipitation (mm)

0–100 100–250 250–500 >500

▲ Palaeoanthropological site between 50 and 500 thousand years old

**Fig. 1 | Palaeoanthropological sites in Southwest Asia dating to between 50 and 500 ka. a**, Sites plotted on a map of modern biomes. **b**, Sites plotted on map of modern precipitation. **c**, Sites plotted on model of MIS 5e precipitation[34], as an illustration of the changes that occurred during humid periods.

a detailed record for the woodlands, a southern extension of the Palae-arctic biome[4,16]. However, in the past decade, research in the Arabian Peninsula has begun to document hominin occupations of the arid Saharo-Arabian biome during episodically wetter periods character-ized by grasslands, lakes and rivers[5,10,17–29]. Emerging patterns of spa-tially divergent cultural-evolutionary developments in Southwest Asia include a young (less than 200 thousand years ago (ka)) Acheulean in central Arabia[24], a technology typically associated with earlier hominins such as *Homo erectus*. There are also repeated manifestations of distinc-tive local characteristics, commonly interpreted as autochthonous developments, in 'refugial' areas in southern Arabia[22,23].

Despite these advances, the few reported sites in interior and north-ern Arabia[20,24–29] (Supplementary Information, section 1) have small sample sizes of artefacts, and are often raw material procurement and workshop sites, with a very different character to the cave and rock-shelter 'living sites' that dominate the Levantine woodland record. The absence of permanent fluvial systems and deeply stratified cave sequences in Arabia has hampered the construction of long-timescale archaeological and hydroclimatic sequences. This has limited efforts to recognize important patterns in archaeological and palaeontological records associated with changes in hominin distribution, demography and behaviour.

Here we report multiple palaeolake sedimentary sequences with associated lithic (stone tool) assemblages and fossil fauna in the Nefud Desert of northern Arabia, representing the first detailed long-timescale record of hominin occupations in Arabia (Figs. 1–3). Khall Amayshan 4 (KAM 4) consists of a series of superimposed lake sequences within a single interdunal basin (Extended Data Figs. 1, 2, Supplementary Information, sections 2, 3). This site, currently unique in Arabia, has preserved a record analogous to the detailed fluvial archives preserved in regions such as northwest Europe. Additionally, we present further evidence for multiple hominin occupations from excavated sites dat-ing to Marine Isotope Stage (MIS) 7 and MIS 5 from the nearby Jubbah palaeolake basin. Together the KAM 4 and Jubbah assemblages show that there were multiple hominin dispersals into Arabia over the last 400,000 years, in association with a unique hydroclimate record.

Each KAM 4 palaeolake deposit is stratigraphically similar, being predominantly composed of massive or finely-laminated carbonate rich marls overlying sands (Extended Data Figure 3, Supplementary Information, section 3). The similarity of these marls, each formed by a discrete lake phase, implies that the palaeoenvironment of KAM 4

was broadly similar during successive humid phases. The sediments are comparable to other palaeolake deposits from the western Nefud Desert[18,20], but are notable for their stratigraphically distinct and super-imposed character, and abundance of associated lithics and fossils. The sediments at the site are fine-grained (sands, silts and marls), reflecting deposition under low-energy or still-water conditions. Larger clasts (gravels) are absent, emphasizing the lack of higher-energy current flow processes feeding the basin during sediment accumulation. Reworking of lithics and fossils from the surrounding landscape into these lake bodies is therefore highly unlikely. Consequently, we argue that the assemblages of lithics and the fossils found in association with these deposits are in situ, which was confirmed by excavations in the case of the Northwest Lake.

The unique KAM 4 record has survived owing to migrating sand dunes that moved in a conveyor-belt-like fashion across the basin, protecting older parts of the sequence from erosion and preventing the mixing of the distinct archaeological and palaeontological assemblages associ-ated with each lake phase. KAM 4 provides the first long-term composite sequence for the later Middle Pleistocene and Late Pleistocene in Ara-bia, with each phase of hominin occupation associated with a broadly similar environment and lithic raw material availability.

The oldest deposit at KAM 4, the Central Lake, is dated by lumines-cence to 412 ± 87 ka (Fig. 2, Extended Data Figs. 2, 3, Supplementary Information, section 5). The Central Lake deposits are also heavily iron-stained compared with other deposits at the site, attesting to their greater antiquity within the basin. The Central Lake is stratigraphically overlain by the southernmost edge of the Northeast Lake, which yielded luminescence ages of 337 ± 39 ka and 306 ± 47 ka. Although the age estimates for both the Central and Northeast Lakes have large uncer-tainties, we emphasize evidence for regional aridity in the millennia either side of both MIS 11 and 9 (Fig. 2), making attribution of Central Lake to MIS 11 and Northeast Lake to MIS 9 parsimonious. The Northwest Lake, which partly overlies the Northeast Lake, is dated by a suite of luminescence estimates on carbonate-rich sands bracketed between two phases of marl deposits to between 192 ± 20 ka and 210 ± 16 ka. A direct U-series age estimate of a bovid fossil from the same layer pro-duced a consistent date of 205 ± 2 ka (2σ) (Supplementary Information, section 6). The Northwest Lake can therefore be correlated with MIS 7, the final humid phase of the Middle Pleistocene. The Southwest Lake has a luminescence estimate of 143 ± 10 ka, and therefore dates either to late MIS 6 or, less probably, to the transition to MIS 5. Sands underlying

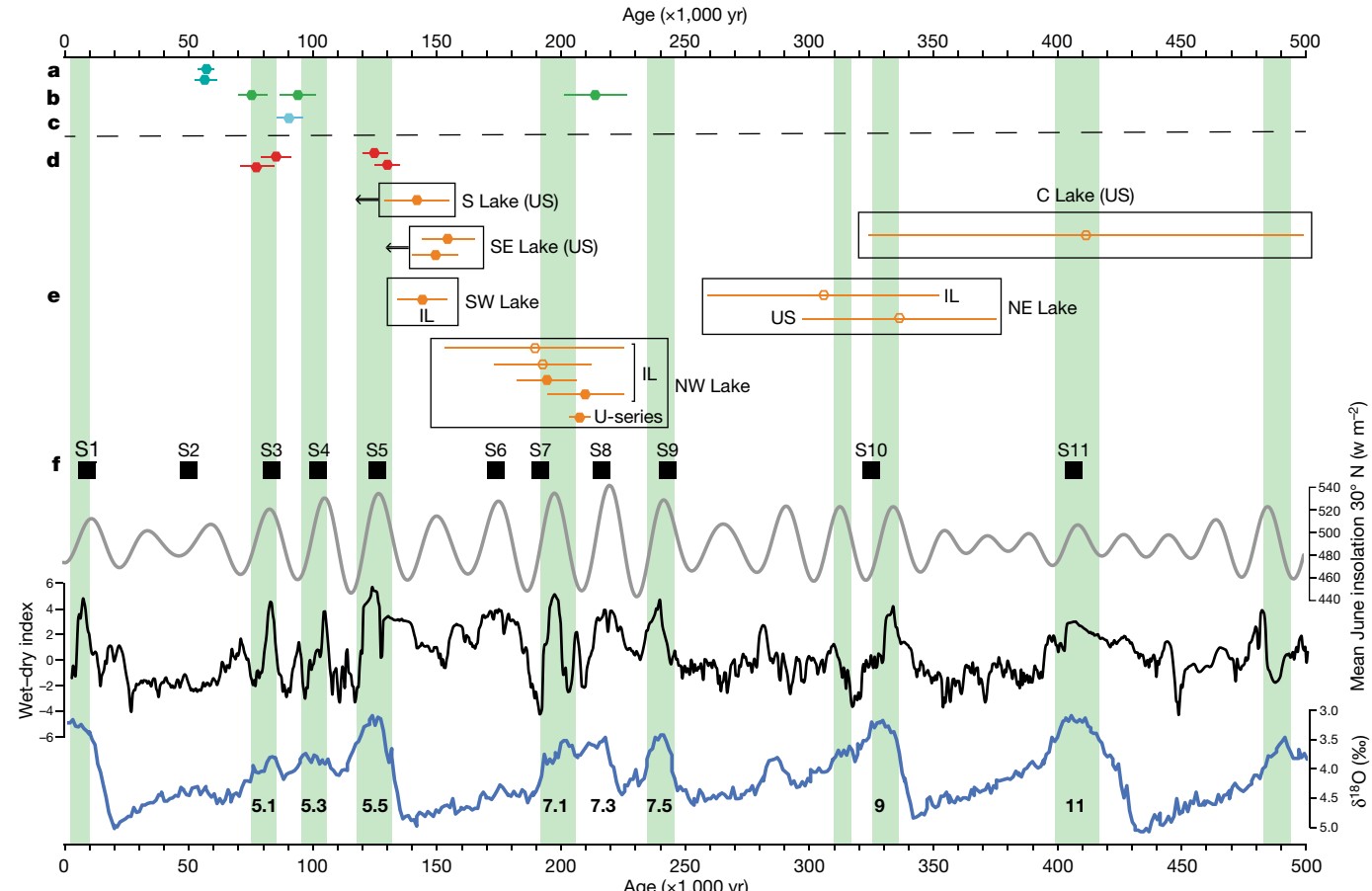

**Fig. 2 | The chronology and environmental context of hominin occupations in northern Arabia. a**, Al Marrat 3[27]. **b**, Jebel-Qattar 1[28]. **c**, Al Wusta[20]. **d**, JSM 1 (present study). **e**, Central (C), Northeast (NE), Northwest (NW), Southwest (SW), Southeast (SE) and South (S) lakes at KAM 4. US, an age for sands underlying the lake (that is, a maximum age for overlying phase of lake formation); IL, in lake (direct date on sediments within lake-related deposits). Black arrows pointing to the left (Southeast and South Lakes) reflect that the luminescence ages provide maximum ages, and the overlying lakes are younger. Filled symbols show quartz ages and open symbols show feldspar ages. **f**, East Mediterranean sapropel record[35], insolation[36] (grey), monsoon index[37] (black) and oxygen isotope record[38] (blue). Southern Arabian humid periods are defined by speleothems in green[21]. Luminescence ages are presented with $1\sigma$ uncertainties and the single U-series age is presented with a $2\sigma$ uncertainty.

the Southeast Lake, which overlies the Central and Northeast Lakes, yielded luminescence estimates of $159 \pm 11$ ka to $149 \pm 9$ ka, similar to the South Lake at $168 \pm 12$ ka to $142 \pm 13$ ka. These maximum age estimates derived from underlying sand indicate that the Southeast and South Lakes date to either the final millennia of the Middle Pleistocene or, more probably, to the subsequent Late Pleistocene. As discussed below, we hypothesize on archaeological grounds that the South Lake dates to MIS 5 and the Southeast Lake to early MIS 3.

The Jubbah record, consisting of excavated stratified lithics, enables us to further extend the occupation sequence of the region. Substantially enlarged excavations at Jebel Qattar 1 (JQ 1) increased the sample size of lithics dating to $211 \pm 16$ ka reported by ref. [28] by 250%. At Jebel Umm Sanman 1 (JSM 1), four new trenches were placed immediately west of earlier test excavations[28]. The JSM 1 trenches revealed deep (more than 1.5–2.5 m) stratigraphic sequences, comprising a series of silty sands with variable frequencies of local gravel clasts. Luminescence dating indicates that the lower part of the JSM 1 sequence dates to $130 \pm 10$ ka, whereas the upper part, in which lithics were found, dates to approximately 75 ka ($77 \pm 7$ ka, $72 \pm 6.4$ ka) (Supplementary Information, section 5).

Each phase of lake formation (apart from the Southwest Lake) at KAM 4 is associated with a distinct lithic assemblage (Fig. 3, Extended Data Figs. 4–8, Supplementary Information, section 7). Assemblage A (Central Lake, approximately 400 ka) consists of handaxes and associated

debitage (Extended Data Fig. 4) and is the oldest dated Acheulean assemblage in Arabia. It shows the production of small and refined handaxes produced by shaping (*façonnage*) of angular slabs of quartzite. Assemblage B (Northeast Lake, approximately 300 ka) is also characterized by the production of small handaxes (Extended Data Figs. 5, 6). These handaxes are rather homogeneous in their technology and morphology, being small and pointed. Core reduction technology to produce flakes is also present in low frequencies in assemblage B, mostly characterized by preferential Levallois reduction. The subsequent assemblage C (Northwest Lake, approximately 200 ka) shows a Middle Palaeolithic technology. Lithics recovered from the surface and from excavations show a complete absence of handaxe manufacture, and a focus on Levallois technology, often centripetal in character (Extended Data Fig. 7), but somewhat diverse (Supplementary Information, section 7). Assemblage D (Southeast Lake, approximately 125–75 ka) and assemblage E (South Lake, approximately 55 ka) are both of Middle Palaeolithic character—assemblage D has a focus on centripetal Levallois technology and assemblage E has a somewhat diverse technology, but with a strong component of unidirectional-convergent preparation to produce convergent Levallois flakes.

With the enlarged excavations at JQ 1, the assemblage dating to approximately 210 ka has a clearly Middle Palaeolithic character; Levallois flakes are present, and bifacial technology is absent (Supplementary Information, section 8). The JSM 1 assemblage from approximately

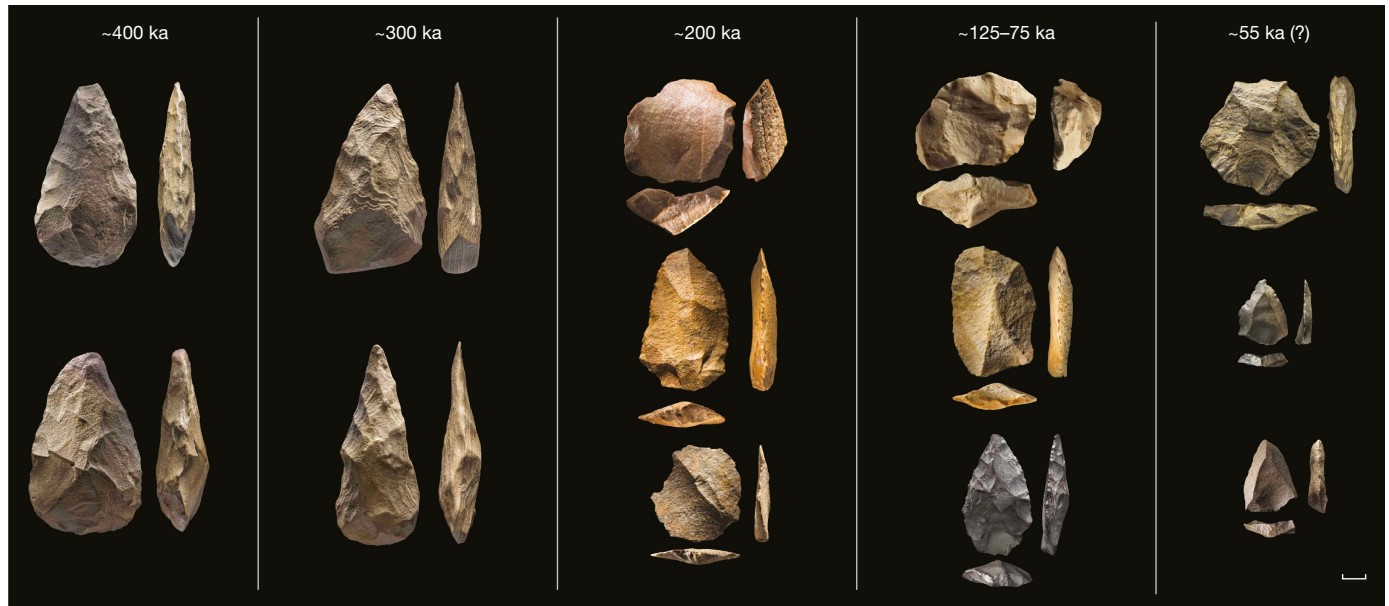

**Fig. 3 | Stone tools from KAM 4 and Jebel Umm Sanman 1 (JSM 1).** From left to right: assemblage A, KAM 4 (approximately 400 ka), assemblage B, KAM 4 (approximately 300 ka), assemblage C, KAM 4 (approximately 200 ka), JSM 1 (approximately 75 ka), assemblage E, KAM 4 (approximately 55 ka?). Scale bar, 1 cm.

75 ka is the largest excavated and dated Palaeolithic assemblage from northern Arabia, and shows a clear focus on centripetal Levallois reduction, with 83% of Levallois flakes having centripetal scar patterns (Extended Data Fig. 9).

This unique record of hydroclimate and associated hominin occupations demonstrates that Acheulean Lower Palaeolithic technology was present during late Middle Pleistocene wet phases, with Levallois technology being present in the final stage of the Acheulean. Assemblages showing similarities to the Acheulo-Yabrudian of the Levantine woodlands have not been identified in Arabia, highlighting distinct trajectories within Southwest Asia. From MIS 7, Arabian Middle Palaeolithic assemblages appear with each phase of increased precipitation, showing varying technological foci in terms of the reduction methods used, from varied Levallois in MIS 7, to centripetal Levallois in MIS 5[3,20], and unidirectional-convergent in MIS 3[23,27].

KAM 4 assemblage C has technological characteristics—such as frequent centripetal Levallois flaking—that are more similar to those of the East African Middle Stone Age (MSA) than to the contemporaneous Levantine early Middle Palaeolithic (Supplementary Information, sections 7, 8). Principal components analysis (PCA) of late Middle Pleistocene Levallois flakes shows that KAM 4 assemblage C falls between Omo Kibish-AHS in East Africa[30] and Misliya in the Levant[31]. For the Late Pleistocene, PCA distinguished between *H. sapiens*-associated assemblages such as Omo Kibish-BNS[30] and Al Wusta in Arabia[20], and the Neanderthal-associated Levantine assemblages from Kebara[15] and Tor Faraj[32]. KAM 4 assemblage D and JSM 1 orientate to the MIS 5 *H. sapiens* assemblages, whereas the negative score on the second principal component for KAM 4 assemblage E orientates it to MIS 4–3 Levantine Neanderthal assemblages.

Animal fossils (primarily vertebrates) from KAM 4 allow us to reconstruct the palaeoenvironmental and biogeographical context of hominin occupations. Hippopotamus fossils had previously been reported in Arabia from MIS 5 contexts (fer example, in refs. [20,33]). KAM 4 shows that hippopotamuses were also present during MIS 7 and, provisionally, MIS 9 (Supplementary Information, section 10). We also identified hippopotamus in the surface scatter of fossils at the nearby site of Ti's al Ghadah. The repeated presence of hippopotamuses, which are obligate semi-aquatic mammals that require permanent water bodies several metres deep, provides powerful evidence for the extent of

environmental amelioration during repeated 'green Arabia' pluvial phases. In addition, the KAM 4 palaeontological assemblages contribute to a growing corpus of evidence indicating that Arabian mammal fauna had stronger affinity with Africa in the Middle and Late Pleistocene than with the Levantine woodland zone[4,33]. The presence of African bovid taxa such as *Syncerus* and *Hippotragus* in northern Arabia indicates the repeated establishment of contiguous regions of grasslands across North Africa and Arabia with abundant freshwater, providing dispersal routes for a variety of species, including hominins. Arabia, however, also features Eurasian and endemic taxa (Supplementary Information 10), indicating that it was a key biogeographical nexus between Africa and the rest of Eurasia that may have also comprised an important interaction zone for hominins.

The northern Arabian late Middle Pleistocene lithic assemblages likewise show greater similarities to African assemblages than to those of Levantine woodland zone sites. The continued production of large handaxes and cleavers in central Arabia at the time the Middle Palaeolithic had appeared in northern Arabia[24] indicates high levels of population structure at this time, perhaps to the extent of different hominin species occupying the region. In MIS 5, it seems that much of Northeast Africa and Southwest Asia shared similar material culture, consistent with widespread dispersals of *H. sapiens*[20]. Subsequently the cooling and aridification of the last glacial cycle led to the fracturing and decline of populations. Renewed dispersals, perhaps including Neanderthals from the north, occurred during the partial amelioration of early MIS 3 (around 59–50 ka). Comparatively stable environmental and ecological conditions in areas such as the Levantine woodland fostered the development of distinctive localized material culture phases[11]. By contrast, the record of interior northern Arabia indicates pulses of occupation during episodic phases of increased environmental humidity, seemingly followed by repeated regional depopulation under increasing aridity.

We have identified at least five pulses of human dispersal into northern Arabia, each associated with a phase of decreased aridity. The differences in material culture between these phases—with two phases of Acheulean technology and then three distinct forms of Middle Palaeolithic—suggests that diverse hominin populations, and probably even species, were expanding into the region at different times (we discuss the implications of our findings further in Supplementary Information,

section 11). The emerging palaeoanthropological record of Arabia highlights the dynamism and regional distinctiveness of Middle and Late Pleistocene hominin demography and behaviour in different parts of Southwest Asia. These processes were intimately connected to regional climatic changes. The available record emphasizes pulsed, long-ranging terrestrial dispersals followed by local variation, and finally population contraction. Given the temporal overlap of radically different technologies within Arabia, and the biogeographical evidence for faunal mixture, it is possible that some of the hominin admixture processes identified by genetic analyses occurred in this region. Arabia, and Southwest Asia more generally, is therefore a key region for unravelling not only the increasingly complex history of how our species spread beyond Africa, but more broadly, how our species' recent success relates to a longer history of hominin dispersals, regional developments and admixture, which occurred in a context of marked environmental oscillation.

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

# Methods

## Site identification and survey

KAM 4 was initially identified through remote sensing analysis[26] (Supplementary Information, section 2). Two main seasons of research were conducted at the site (2014 and 2017) as part of the Palaeodeserts/Green Arabia Project. The site was systematically surveyed with pedestrian transects. Using a total station and Trimble XRS Pro Differential Global Positioning System, the topography of the site was recorded in detail, and all points of interest (stone tools, fossils and sedimentary features) were recorded and entered into a geographic information system. JSM 1 and JQ 1 were first identified in 2011[28]. We carried out renewed excavation of the sites in 2013. With JQ 1, the stratigraphic sequence was already understood, so the aim was simply to increase the sample size of lithics. At JSM 1, the original excavations had rapidly hit bedrock, so renewed excavations were conducted slightly further west in the hope of identifying deeper stratigraphic sequences.

## Stratigraphy and sedimentology

Sections for sedimentary analysis and luminescence sampling were excavated for each of the palaeolake phases at KAM 4 (Supplementary Information, section 3). The Northwest Lake was identified as having the best potential to recover buried material, as fossils and lithics appeared to be emerging from sediments, so four trenches (1–4) were dug here. These trenches and the excavations at JSM 1 and JQ 1 (Supplementary Information, section 4) were conducted using single-context excavation methods. All sediments were dry sieved through 5-mm mesh. The focus of this paper is on the archaeological assemblages, and not detailed palaeoenvironmental analysis so our sedimentary description consists of field observations from logging sections. Fossils from KAM 4 have previously been reported[33].

## Chronometric dating methods

We used luminescence (OSL on quartz and pIRIR on feldspar) methods to date the sedimentary deposits at KAM 4 and at JSM 1 (Supplementary Information, section 5). These measure the time since sediments were last exposed to sunlight. Opaque metal tubes were hammered into cleaned sections, transported to the UK and analysed as described in Supplementary Information, section 5. Environmental dose rates were calculated using location and overburden density (cosmic rays), field gamma spectrometry (gamma), and thick-source beta counting (beta). A bovid tooth (KAM16/85) was recovered from unit 3 of the Northwest Lake at KAM 4. A direct age was obtained using the U-series dating method, which dates the moment uranium is incorporated into the fossil. Powdered samples of both enamel and dentine tissues were drilled from the tooth at Griffith University, and U-series analyses were subsequently carried out at the University of Queensland. While it was initially planned to combine with electron spin resonance analyses, the U-series results obtained showed that the tooth was not suitable for that purpose (Supplementary Information, section 6).

## Lithic analysis

Lithics (stone tools) from the excavations at all sites and from the systematic transect survey at KAM 4 were studied using the methodology described previously in refs. [9,20] and in Supplementary Information, sections 7, 8. Our initial focus was on describing the basic typo-technological features of the assemblages. We selected illustrative examples for photography, 3D scanning, and illustration. For the Middle Palaeolithic samples, we carried out a full metric and attribute analysis following the above references and references therein. As well as allowing the description of the assemblages in quantitative terms, we focussed on the characteristics of Levallois flakes from these assemblages as a way to compare assemblages. We did this both in terms of univariate features (such as dorsal scar patterns), and using PCA to compare the morphology of Levallois flakes between the assemblages (Supplementary Information, section 8).

## Reporting summary

Further information on research design is available in the Nature Research Reporting Summary linked to this paper.

## Data and code availability

Data for the PCA analysis are archived at https://doi.org/10.5281/zenodo.5082293. All other relevant data are included in the paper and Supplementary Information.

## Code availability

Code for the PCA analysis are archived at https://doi.org/10.5281/zenodo.5082293.

**Acknowledgements** We thank the Heritage Commission, Ministry of Culture, Saudi Arabia for fieldwork support and permission to conduct this research. The research was funded by the Max Planck Society, the European Research Council (295719 to M.D.P.), the British Academy (H.S.G.), the Leverhulme Trust (ECF-2019-538 to S.B., ECF-2019-538 to P.S.B. and PG-2017-087 to S.B., E.A., S.J.A. and M.D.P.), the Research Council of Norway, through its Centres of Excellence funding scheme, SFF Centre for Early Sapiens Behaviour (SapienCE), project number 262618 (S.J.A.), the Natural Environmental Research Council (NERC) through its London DTP studentship funding (R.C.-W.), The Nature and Science Researchers Supporting Project (NSRSP-2021-5), DSFP, King Saud University, Riyadh, Saudi Arabia (A.M.A.), the Leakey Foundation (M.S.) the Australian Research Council (FT160100450 to J.L. and FT150100215 to M.D.), and the Spanish Ramón y Cajal Fellowship (RYC2018-025221-I to M.D.). We thank L. Clark-Balzan for assistance with the luminescence dating, I. Cartwright for lithic photography and M. O'Reilly for assistance with figures. We thank the museums listed in the Supplementary Information for access to comparative collections.

**Author contributions** Conceptualization: H.S.G., A.M.A. and M.D.P. Data collection: all authors. Luminescence dating: E.A. and S.J.A. U-series dating: G.J.P. and M.D. Palaeoenvironmental analysis: T.S.W., E.A., R.C.-W., P.S.B., S.J.A., N.D., J.L., A.P. and I.C. Lithic analysis: H.S.G., E.M.L.S. and J.B. PCA: W.C.C. Writing: all authors.

**Funding** Open access funding provided by Max Planck Society.

**Competing interests** The authors declare no competing interests.

**Additional information**
**Correspondence and requests for materials** should be addressed to H.S.G. or M.D.P.

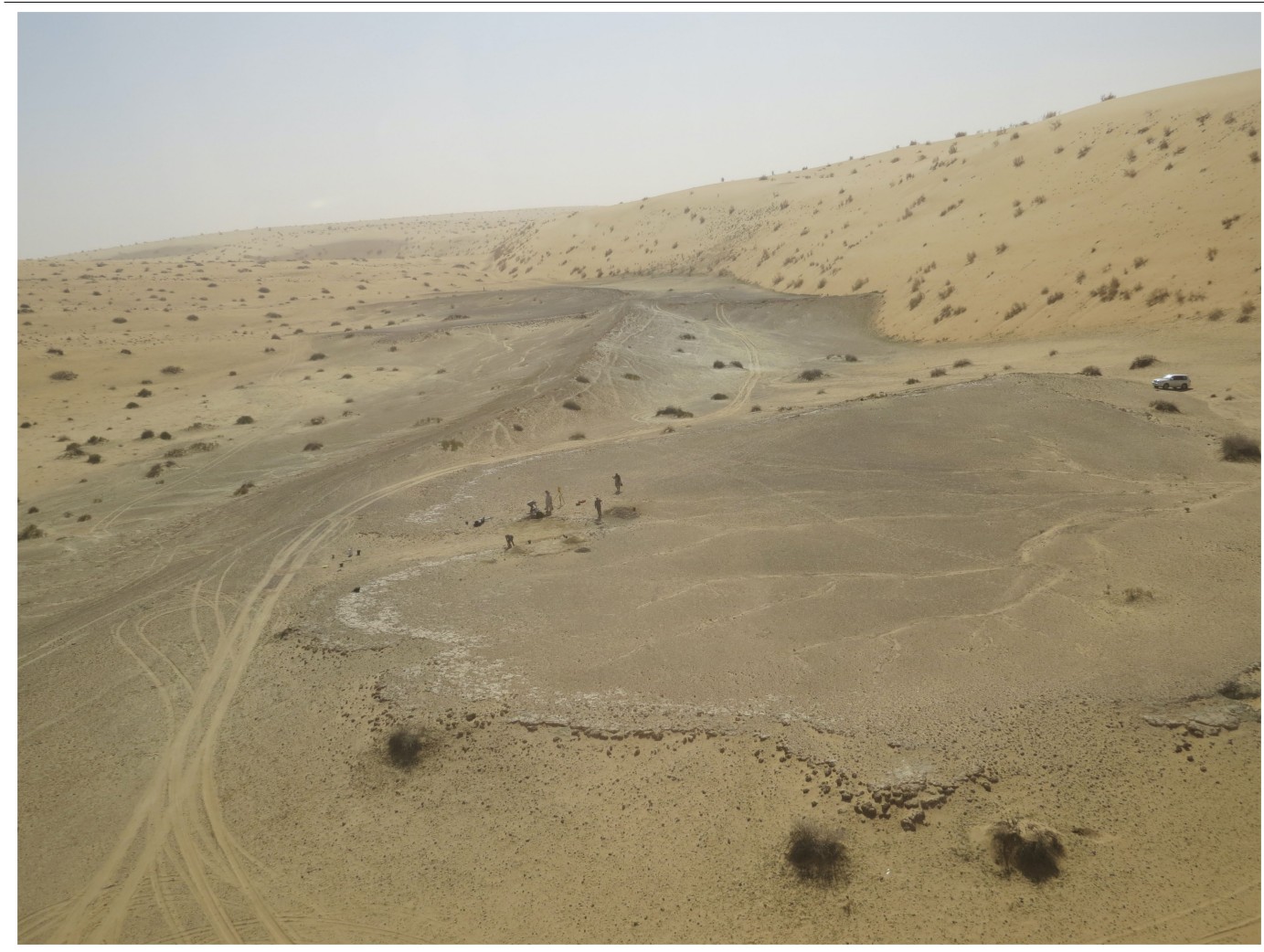

**Extended Data Fig. 1 | KAM 4 looking south, team members are visible on the Northwest Lake.**

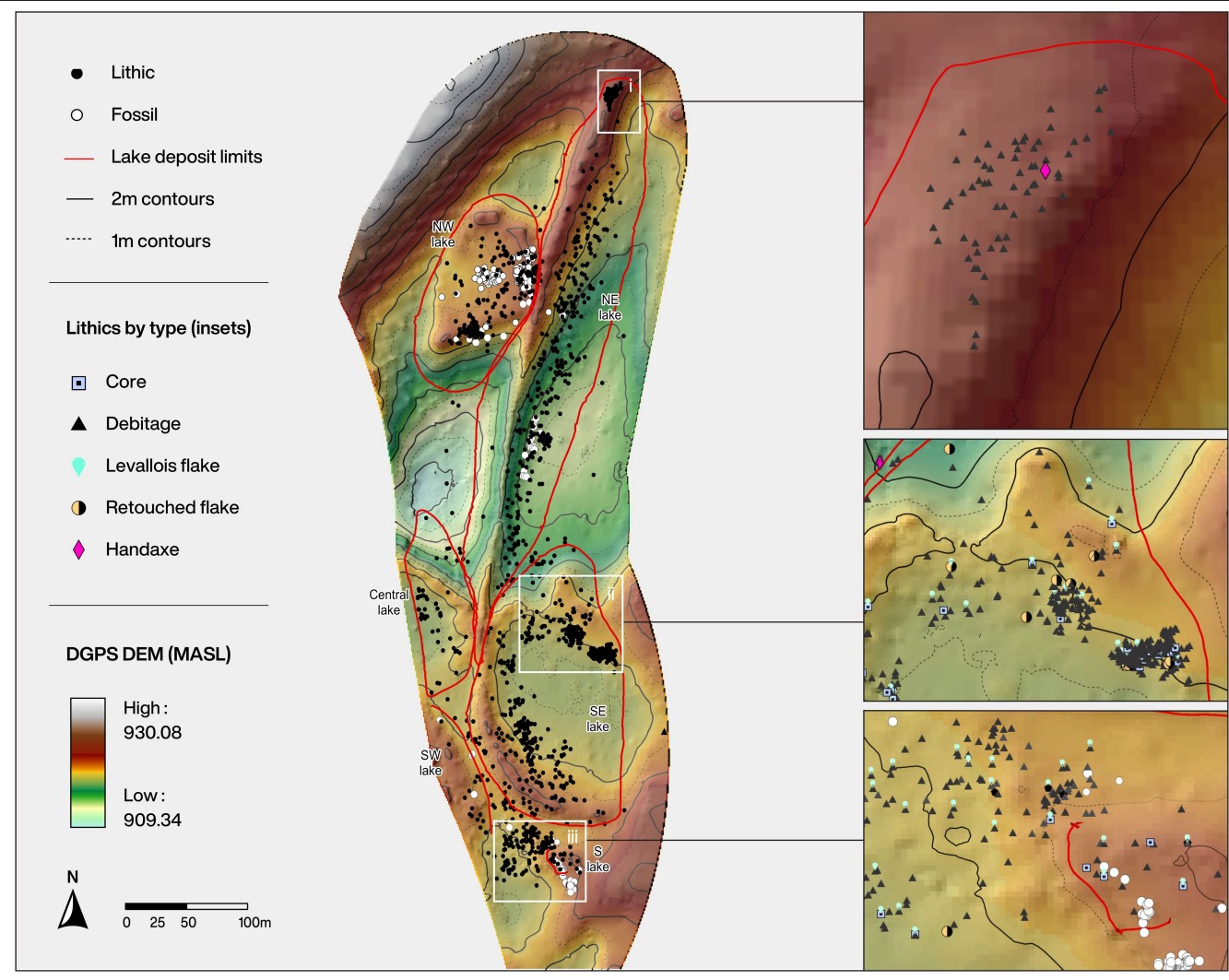

**Extended Data Fig. 2 | Plan of Khall-Amayshan-4, showing multiple phases of lake formation, associated with lithic and fossil assemblages.**

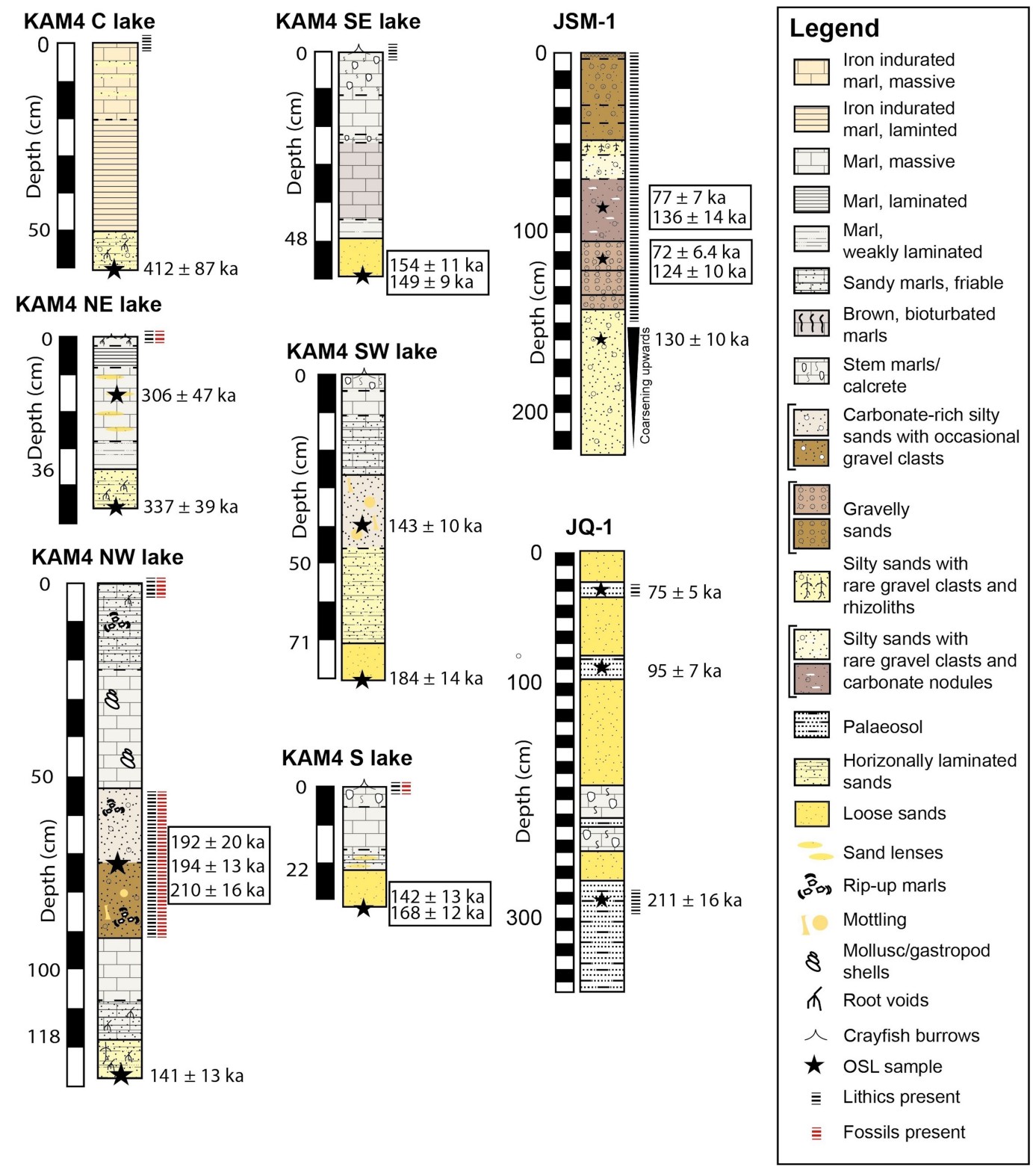

**Extended Data Fig. 3 | Stratigraphic logs and chronometric dates for KAM4, JSM1 and JQ1.**

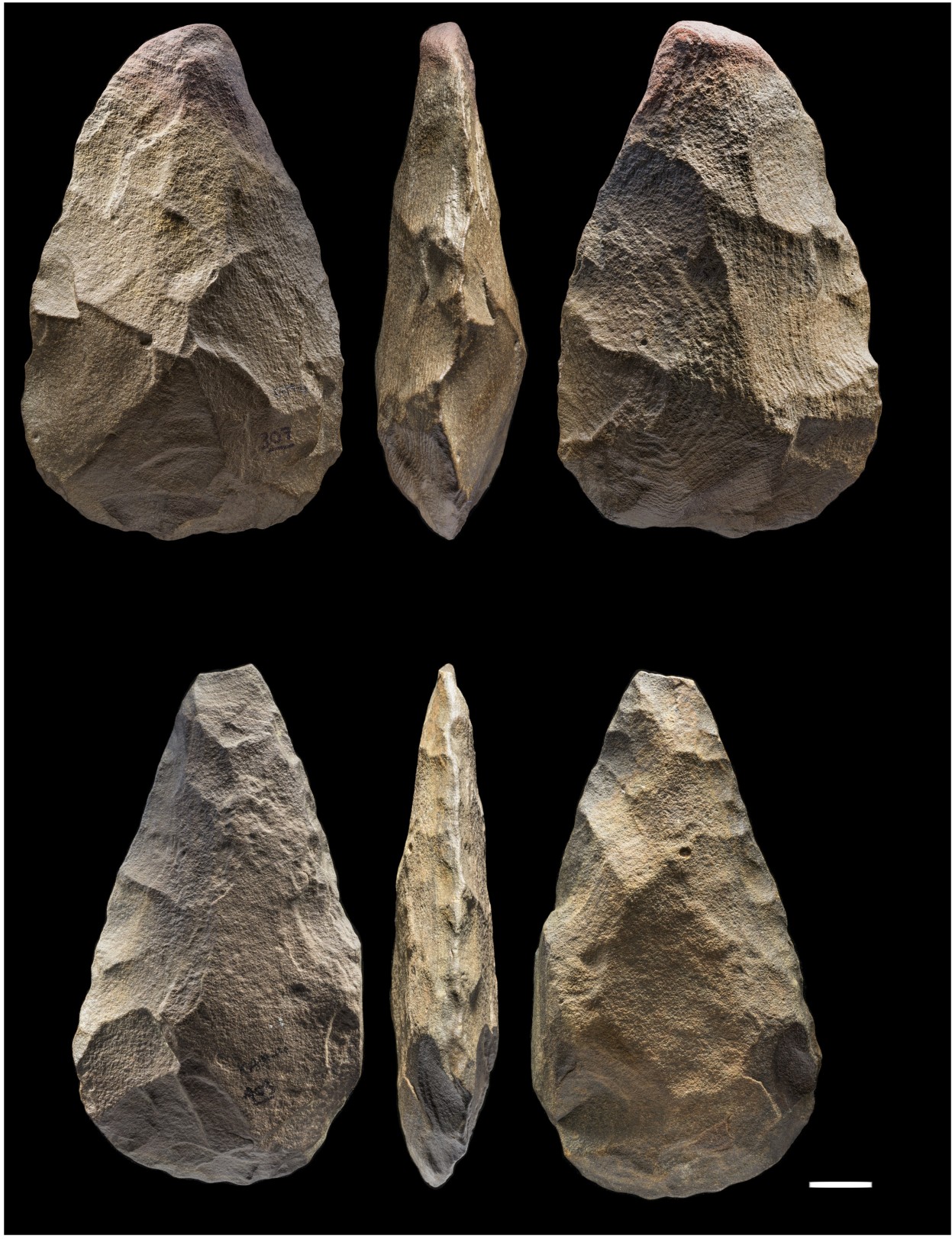

**Extended Data Fig. 4 | Assemblage A handaxes from Central Lake (MIS 11), KAM 4.** Scale: 1 cm.

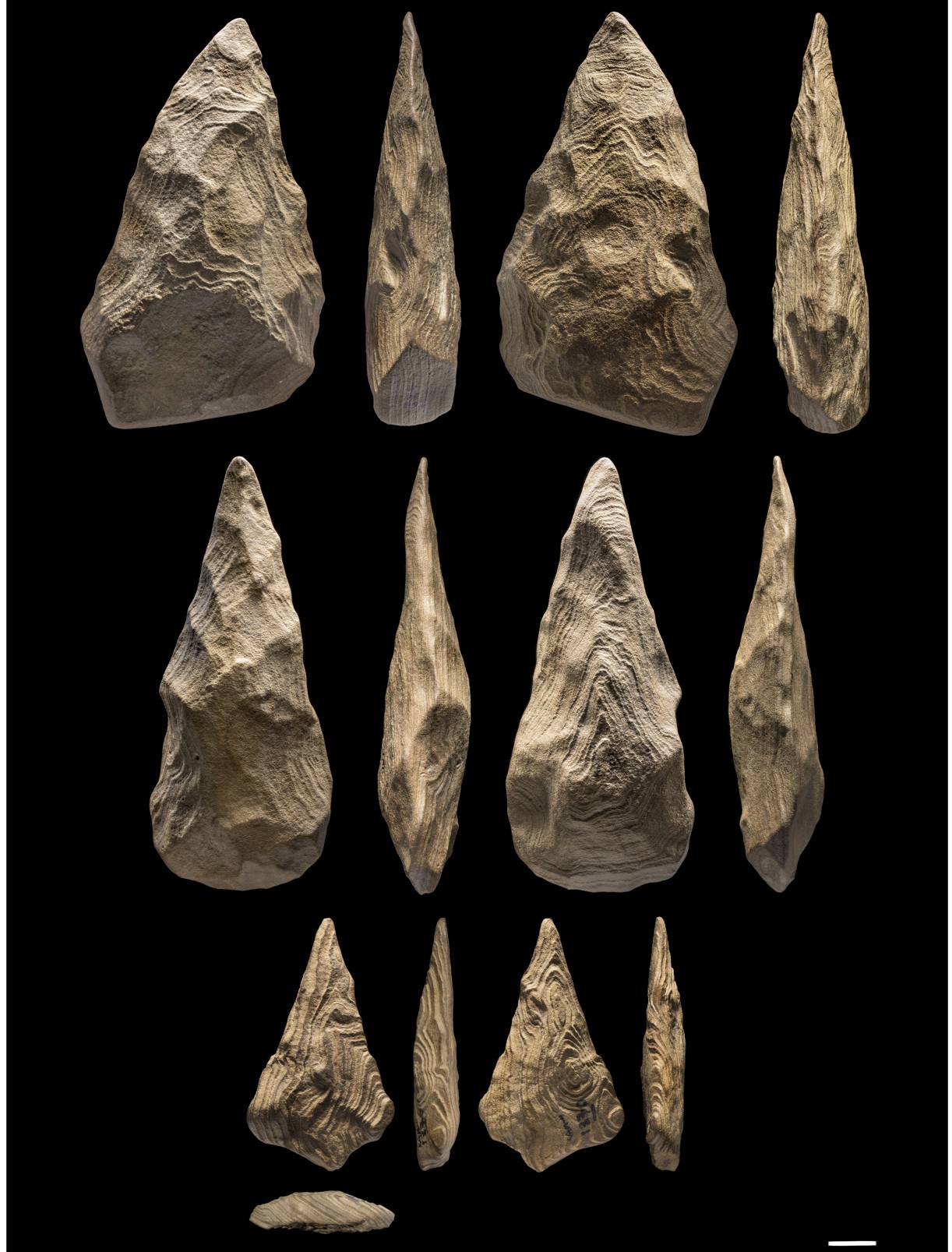

**Extended Data Fig. 5 | Assemblage B handaxes from Northeast Lake (MIS 9), KAM 4.** Scale: 1 cm.

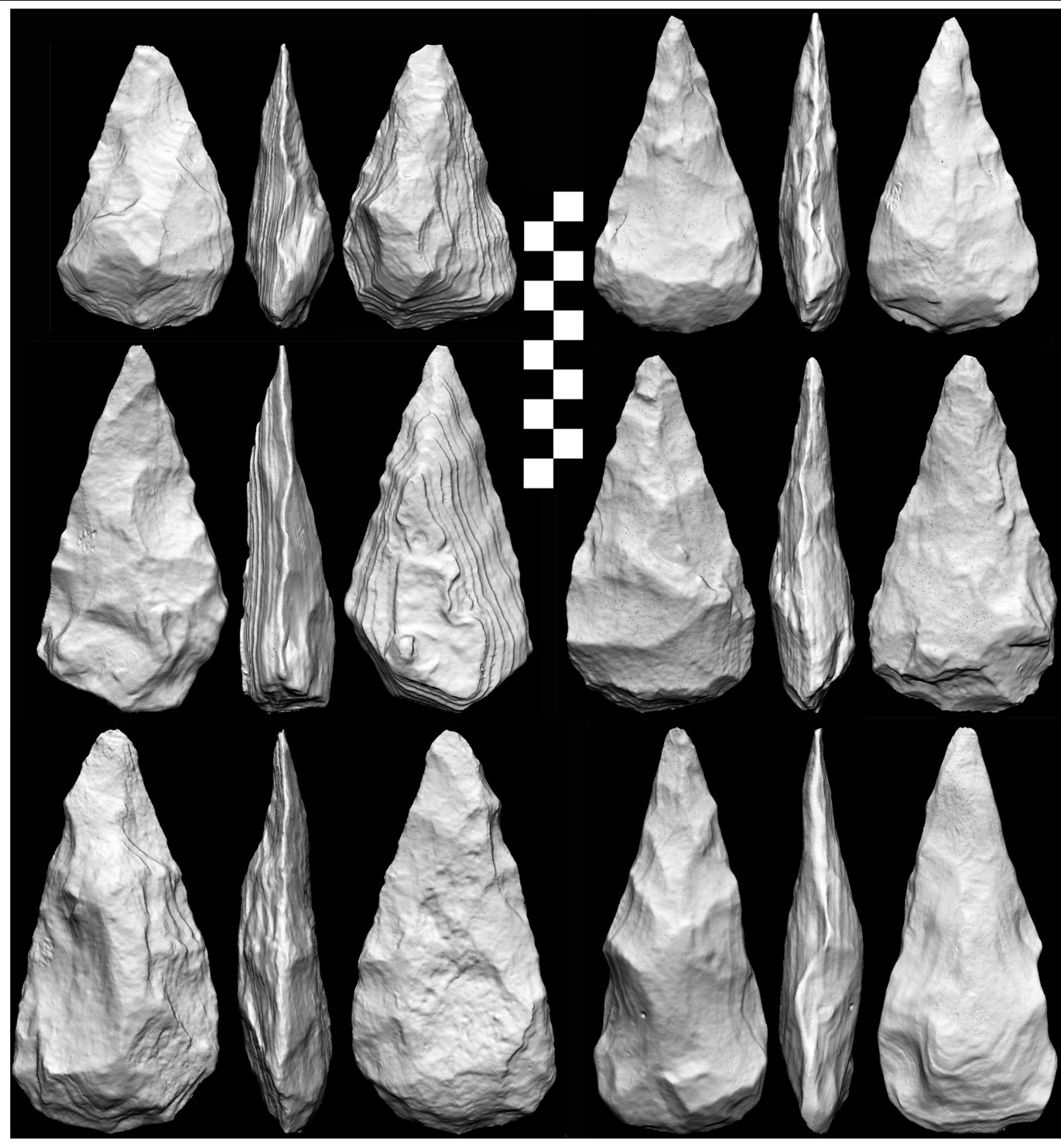

**Extended Data Fig. 6 | 3D scans of assemblage B (MIS 9) handaxes from KAM 4.**

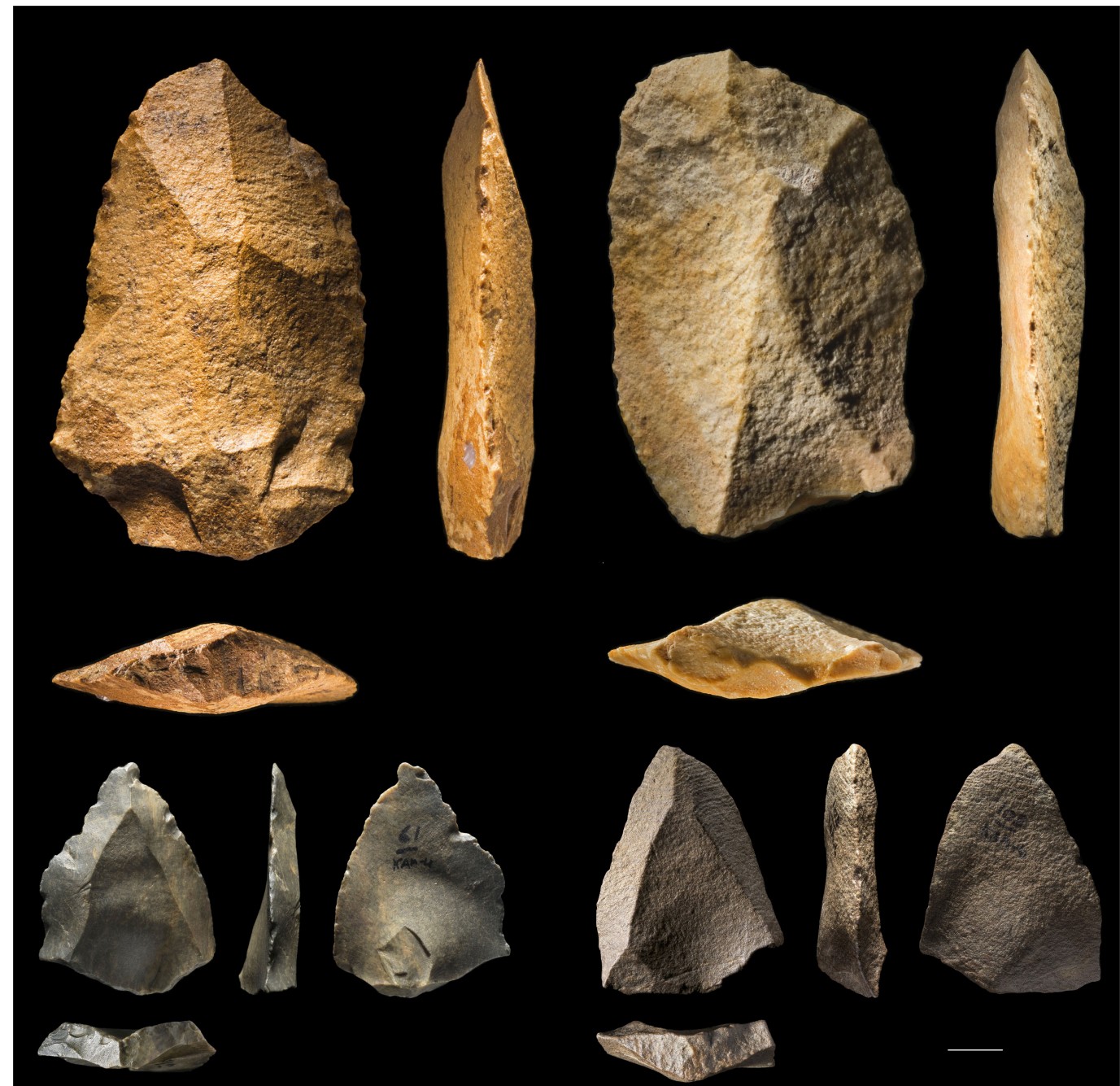

**Extended Data Fig. 7 | Middle Palaeolithic artefacts from KAM 4 and JSM 1.** Top left: KAM 4 assemblage C (MIS 7) Levallois flake, top right: JSM 1 Levallois flake (MIS 5), bottom row: Levallois points from assemblage E (MIS 3?), KAM 4. Scale: 1 cm.

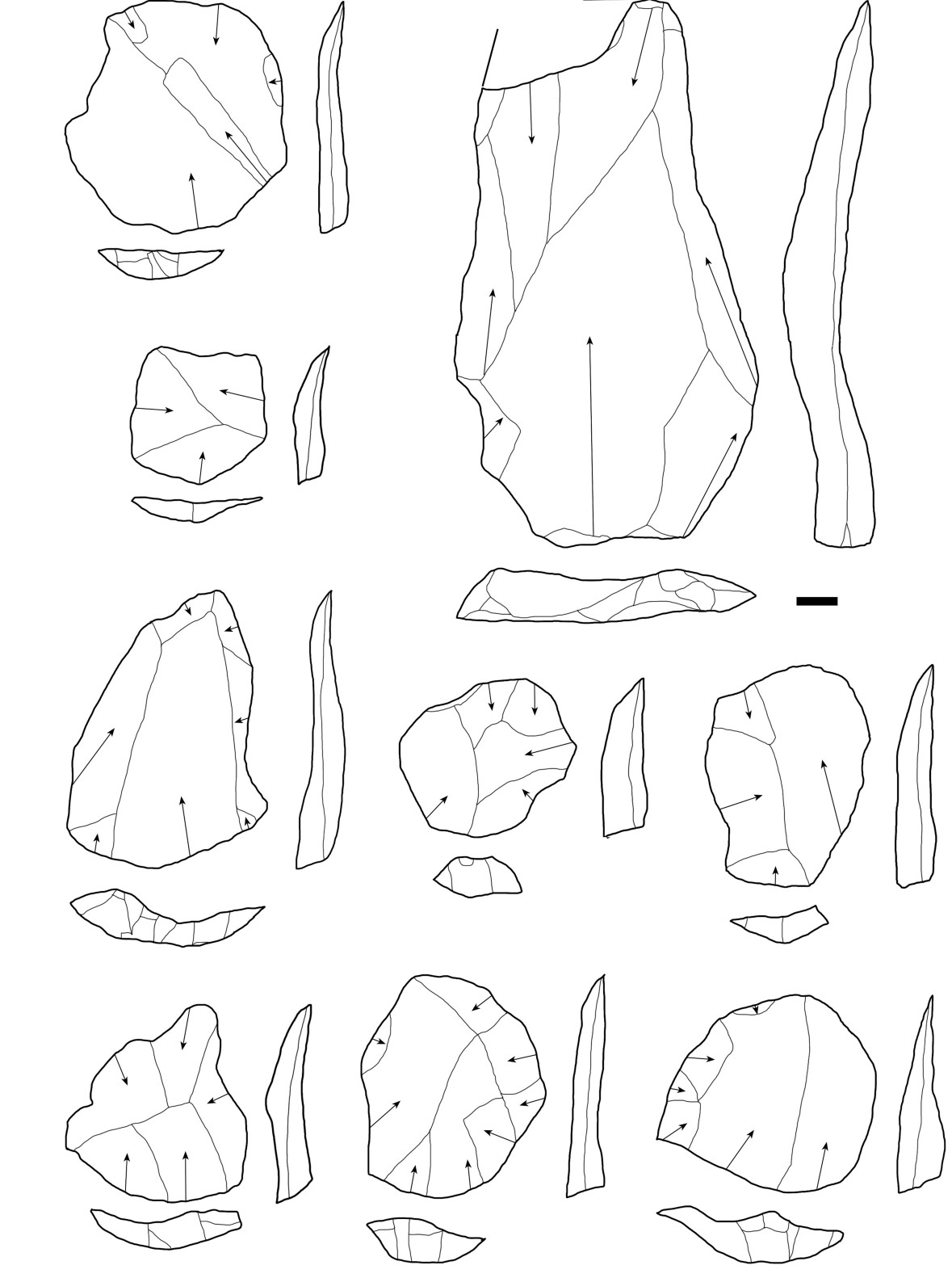

**Extended Data Fig. 8 | Levallois flakes from assemblage C, KAM 4 (MIS 7).** Scale: 1 cm.

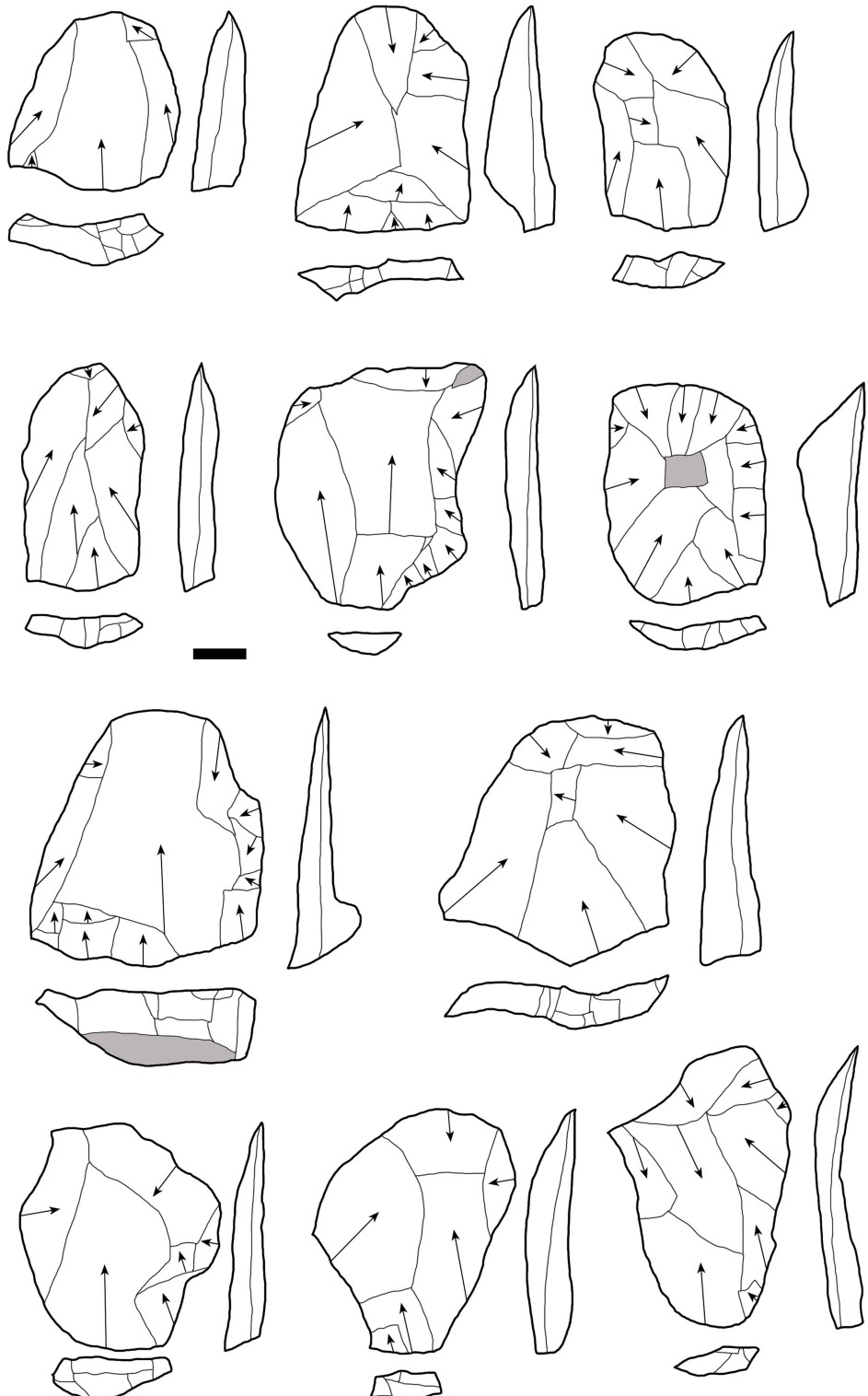

**Extended Data Fig. 9 | Levallois flakes from JSM 1 (MIS 5, ca. 75 ka).** Scale: 1 cm.

# Reporting Summary

## Statistics

For all statistical analyses, confirm that the following items are present in the figure legend, table legend, main text, or Methods section.

| n/a | Confirmed | |
|---|---|---|
| ☐ | ☒ | The exact sample size ($n$) for each experimental group/condition, given as a discrete number and unit of measurement |
| ☐ | ☒ | A statement on whether measurements were taken from distinct samples or whether the same sample was measured repeatedly |
| ☐ | ☒ | The statistical test(s) used AND whether they are one- or two-sided *Only common tests should be described solely by name; describe more complex techniques in the Methods section.* |
| ☒ | ☐ | A description of all covariates tested |
| ☐ | ☒ | A description of any assumptions or corrections, such as tests of normality and adjustment for multiple comparisons |
| ☐ | ☒ | A full description of the statistical parameters including central tendency (e.g. means) or other basic estimates (e.g. regression coefficient) AND variation (e.g. standard deviation) or associated estimates of uncertainty (e.g. confidence intervals) |
| ☐ | ☒ | For null hypothesis testing, the test statistic (e.g. $F$, $t$, $r$) with confidence intervals, effect sizes, degrees of freedom and $P$ value noted *Give P values as exact values whenever suitable.* |
| ☒ | ☐ | For Bayesian analysis, information on the choice of priors and Markov chain Monte Carlo settings |
| ☒ | ☐ | For hierarchical and complex designs, identification of the appropriate level for tests and full reporting of outcomes |
| ☐ | ☒ | Estimates of effect sizes (e.g. Cohen's $d$, Pearson's $r$), indicating how they were calculated |

*Our web collection on statistics for biologists contains articles on many of the points above.*

## Software and code

Policy information about availability of computer code

| | |
|---|---|
| Data collection | For collection of the luminescence data, software specific to the instruments (Risø TL-DA-15 and TL-DA-20) was used. |
| Data analysis | Luminescence data were analysed using Analyst v.4.31.9 (individual equivalent doses and fading rates) and R v. 3.6.1 using v.0.9.10 of the Luminescence package (age models, fast ratios, fading corrections, cosmic dose rates, Abanico plots). The PCA was conducted using R 4.1.0, psych 2.1.3, and ggplot2 package (3.3.3). |

For manuscripts utilizing custom algorithms or software that are central to the research but not yet described in published literature, software must be made available to editors and reviewers. We strongly encourage code deposition in a community repository (e.g. GitHub). See the Nature Portfolio guidelines for submitting code & software for further information.

## Data

Policy information about availability of data

All manuscripts must include a data availability statement. This statement should provide the following information, where applicable:
- Accession codes, unique identifiers, or web links for publicly available datasets
- A description of any restrictions on data availability
- For clinical datasets or third party data, please ensure that the statement adheres to our policy

All relevant data is included in the paper and SI, or for the PCA analysis the data and code is archived at DOI: 10.5281/zenodo.5082293.

# Field-specific reporting

Please select the one below that is the best fit for your research. If you are not sure, read the appropriate sections before making your selection.

☐ Life sciences ☐ Behavioural & social sciences ☒ Ecological, evolutionary & environmental sciences

For a reference copy of the document with all sections, see nature.com/documents/nr-reporting-summary-flat.pdf

# Ecological, evolutionary & environmental sciences study design

All studies must disclose on these points even when the disclosure is negative.

| | |
|---|---|
| Study description | Stone tools and samples for chronometric dating were collected from archaeological sites in the western Nefud Desert of Saudi Arabia. Stone tools were described both qualitatively and with descriptive statistics. For the comparison of Levallois flakes, the structure of the dataset was explored using PCA analysis of standard metric measurements, with comparative material selected from neighbouring regions. No experimental factors or control groups were used in such analyses, where PCA was conducted on the dataset and then data were plotted by assemblage name. Luminescence dating was conducted to determine the ages of the assemblages. |
| Research sample | The locations of study sites were determined by a combination of remote sensing and field prospecting. Systematic pedestrian transects were conducted across the sites and all objects of interest (lithics, fossils, etc.) were recorded using a differential GPS system or total station. Excavations following standard archaeological practices were conducted in areas of particular interest (i.e. with high artefact densities). Sections of trenches were samples for luminescence dating and palaeoenvironmental analysis. For the PCA analysis comparing Levallois flake morphologies, a series of assemblages from surrounding regions were sampled. |
| Sampling strategy | For the comparative lithic aspect, for large assemblages, Levallois flakes were sampled to give sample sizes of several dozen. Sufficient sample sizes were judged as being larger than the number of pieces included in complete assemblages. |
| Data collection | The main forms of data collected for this interdisciplinary study consist of 1) lithic artefacts, which were studied by Huw Groucutt, Eleanor Scerri, and James Blinkhorn. Standard metrics and technological classifications were recorded using goniometers and entered into excel files, 2) Luminescence dating was carried out by Eric Andrieux, Simon Armitage and Richard Clark-Wilson, with input from Laine Clark-Balzan; Gamma dose rates for samples prefixed "PD" were measured using an EG&G Ortec digiDart-LF instrument while for those prefixed "JSM" or "KAM4-OSL", an Inspector 1000 was used. Luminescence measurements were performed using Risø TL/OSL-DA-15 or Risø TL/OSL-DA-20 instruments. Beta dose rates for "JSM" and "PD" samples were measured using a Risø GM-25-5 low-level beta counting system, 3) U-Series dating was conducted by Gilbert Price and Mathieu Duval. Drilling was conducted at Griffith University, and U-Series dating at the University of Queensland |
| Timing and spatial scale | Fieldwork and initial data collected during field seasons in Saudi Arabia in 2013 to 2015. All analyses were conducted in an intermittent manner from 2013 onwards and were completed in 2020. |
| Data exclusions | All data collected were analysed and contributed to the final conclusions |
| Reproducibility | The archaeological analyses are descriptive, and reproducibility is possible by re-analysis of the data. The luminescence component of this study involved measurements of multiple individual quartz and K-feldspar grains, or multi-grain aliquots of the same, yielding distributions of equivalent-dose estimates from which the weighted mean was calculated using well-established statistical models (see SI). |
| Randomization | The only aspect this applies to is a few cases where Levallois flakes were sampled from large assemblages. A random sample of flakes laid out on a table was taken. These data were used for descriptive statistics, categorised by the relevant assemblage. |
| Blinding | No blinding was conducted as we are reporting descriptive statistics, not experimental results. |

Did the study involve field work? ☒ Yes ☐ No

# Field work, collection and transport

| | |
|---|---|
| Field conditions | Fieldwork was conducted in warm (typically in the 20s C) and dry conditions in Saudi Arabia in two main fields seasons. |
| Location | The sites are located in the western Nefud Desert, in northern Saudi Arabia. Supplementary table S1 gives more information. |
| Access & import/export | Permission to conduct the research and relevant export permits were provided by the Saudi Ministry of Culture. Multiple permits were provided to M.D. Petraglia from 2010 onwards. |
| Disturbance | Excavations were backfilled |

# Reporting for specific materials, systems and methods

We require information from authors about some types of materials, experimental systems and methods used in many studies. Here, indicate whether each material, system or method listed is relevant to your study. If you are not sure if a list item applies to your research, read the appropriate section before selecting a response.

## Materials & experimental systems

| n/a | Involved in the study |
|-----|------------------------|
| ☒ | ☐ Antibodies |
| ☒ | ☐ Eukaryotic cell lines |
| ☐ | ☒ Palaeontology and archaeology |
| ☒ | ☐ Animals and other organisms |
| ☒ | ☐ Human research participants |
| ☒ | ☐ Clinical data |
| ☒ | ☐ Dual use research of concern |

## Methods

| n/a | Involved in the study |
|-----|------------------------|
| ☒ | ☐ ChIP-seq |
| ☒ | ☐ Flow cytometry |
| ☒ | ☐ MRI-based neuroimaging |

## Palaeontology and Archaeology

| | |
|---|---|
| Specimen provenance | Fieldwork was conducted in the western Nefud, with permits for research and export issued to M.D. Petraglia from 2010 onwards. |
| Specimen deposition | Archaeological samples are stored at the National Museum, Riyadh, Saudi Arabia. |
| Dating methods | New age estimates were generated by luminescence dating and U-series dating, as discussed above. |

☒ Tick this box to confirm that the raw and calibrated dates are available in the paper or in Supplementary Information.

| | |
|---|---|
| Ethics oversight | All fieldwork was permitted and overseen by the Saudi Ministry of Culture. |

Note that full information on the approval of the study protocol must also be provided in the manuscript.

