## [Peer Review File · Nature]

Manuscript Title: Multiple hominin dispersals into Southwest Asia over the last 400,000 years

Reviewer Comments & Author Rebuttals

Reviewer Reports on the Initial Version:

Referees' comments:

Referee #1 (Remarks to the Author):

The subject matter of this paper is of great and broad interest focussing as it does on trying to bridge the gap between the African Archaeological record where human dispersals are thought to have occurred to Levant and Asian records where people migrated to. To do this the authors have combined a wide range of lake evidence with associated lithic assemblages from the Arabian peninsular. The underlying evidence is fascinating and compelling and of great potential significant and the conclusions appear robust and exciting.

As detailed in comment below I do have some concerns about the chosen 'green' windows.

Abstract: Windows not ranges. Some in huge interglacials other e.g. 55 ka very brief green periods...

P3 L4/5 Saharo-Arabian and Paleoartic biomes implies a north-south migratory dynamic but the paper is more about west-east migrations? Especially as next sentence is arguing for the importance of Asia in Neanderthal-sapiens mixing.

Intro I felt could be more focussed. The key argument is there are thought to have been dispersals and in and out of Africa but evidence of the pathways between sources and destinations has been hampered but the paucity and fragmentary nature of evidence in two key areas- the Sahara for north-south migrations and The arabian peninsula for east-west migrations.

P4 L1 should it be Achuelean culture not just Achuelean?

P4 Fig 1 – "A" could be improved but showing PE not just P as this is of more relevance for the existence and persistence of surface water. Would also benefit either for inset showing where in world fig is of and/or some place names to aid readers locate the area easily. Grey areas are sea but could also be made clearer.

P5 l7/8 refer interchangeably to sequences and formations. They are not the same thing and could be rephrased to aid readability.

P5 l17 would be helpful if SI3 also was annotated to indicate the carbonate rich marls

P5 l22 – I don't quite follow the argument that because the sediments lack coarser material the archaeology is in situ. Probably just needs rephrasing to be clearer.

P6 l6 late Quaternary is being used as a specific time period so should therefore be "the Late Quaternary".

P6 l8-24. The errors on the luminescence ages are large (+/-87 thousand years for the oldest) so I find it unclear how attribution to very precise green windows to 400 300 etc can be warranted.

This needs explanation.

Along these lines explanation is required as to why attribution of Lake events places them in interglacials (e.g MIS11,9,7) when clearly the current interglacial is hyperarid and not conducive to lake formation. Heavy reliance is apparently made on the wet-dry index which is of North African/Mediterranean origin so not regional. Central and NW Lake could have been ascribed to MIS 10 and 8 respectively or even MIS 12 and 10. Later lakes appear to be associated with cooler MIS stage 6. If these lakes were truly short-lived windows of opportunity then understanding PE through time is important as wet periods when temperatures in the Arabian peninsular were also extreme would not allow lake formation but wet periods during cooling episodes would.

P10 I22 refrain from stating "perhaps related to the ca. 300-500 ka sediments. Or to stratigraphically underlying deposits" – its speculative and a point is still made that you found hippo remains at the site.

P10 I24 "iconic evidence" phrasing not justified.

P10 Not my area of specialism but the use of component analysis to tease out affinities looks strong and important for the paper – could it be included somehow in Fig 4 rather than in SI

P12 I1 Unclear how these phased of decreased aridity have been identified or what they are by this point in the paper. Longer-term records of aridity in the peninsular like that of Pruesser et al. or off-shore cores have not been drawn upon. Evidence of the material cultural differences between phases I am guessing is presented in Fig 4 but could be highlighted more.

Ex Fig 3 – shows good agreement between U and luminescence dating techniques and good increase in age with depth for most sites. Its unclear why some ages are boxed with two ages in which do not agree in some instances particularly jsm-1. Some symbology to get across which age is from what technique would be helpful.

Its unclear for figures or text which ages are based on feldspar/IRSL and which temperature and which quartz/OSL. Also in SI no indication was made if a residual dose was subtracted from what is quite a hard to bleach pIRSL290 or pIRSL 225 signals.

Ex fig 4-5 beautifully taken photos!

Referee #2 (Remarks to the Author):

This is a remarkably important paper that addresses an equally important question: What was the role of SW Asia in the dispersal of homies out of Africa. As the authors note, perhaps a bit too emphatically, defensively, the traditional narrative has been that hominins remained in Africa sensu stricto until ~70 ka and burst out, traversed across Arabia and the Levant and populated Europe, SE Asia, and the rest of the world.

New research by this group and others has refined that story to now tell a more inclusive story that hominins had earlier forays into Arabia extending at least back to MI Stage 5 and probably earlier. The SW Asia and European fossil, lithic, and genomic data still hold a firm hold on the earliest old world dispersal around 70 ka, but it's clear hominins were in Arabia way before this during co-called Green phases, when orbital forcing of the monsoon favored wetter, vegetated landscapes suitable for habitation.

This paper is important as it extends this Arabian habitation story back to at least 400 ka, much

earlier than previously appreciated. It's an important result and extremely well documented in terms of the comprehensive geology, dating, archeology, and context of the new results. It's an important addition to the large narrative and worthy of publication in my view.

Aside from the very thorough field documentation (in the 100+ page Supp Mat section), the entire story rests upon thermoluminescence dating of sands containing the artifacts. This method has many uncertainties, and the method is less useful for >300 ka dating, hence the very large 20% uncertainty of the 400 Ka basal occupation site. That said, they fortunately have one tooth where an independent U-series date reveals an age consistent with the thermo dates, and this lends considerable support to the overall study (they not doubt rejoiced). I would like to have seen/see more dates on teeth if they were conducted. Moreover, the lake phases are roughly consistent with known wet phases from the eastern Med and regional speleothem data.

Altogether, this is a very tight story and I have little to critique on the science.

I do think their supposition that the 'northern (Levant" vs. southern" (Arabia) route differentiation implies some fundamentally different mode of human migration in response to environmental change is misguided. During these wet phases the hyperacid desert regions were vegetated with large permanent lakes (resources) and thus were corridors for exchange as was the Levant. So, the point to be made with this paper is rather that the Arabian peninsula was an equivalent migration corridor during wet phases - something we've know for decades (McClure, 1976). The current paper puts people into this narrative, with water, dates, multiple times, and very early on. (Note the authors can include early Holocene (6-9 ka) and late Pleistocene (25-30 ka) Lake occurrences on their Fig. 2)

Bravo !

Referee #3 (Remarks to the Author):

I have no criticism over the contents of the paper, which collates a large amount of excavation and survey. This is an excellent paper by a long-established high-quality team that has already published several papers on the palaeolithic and Pleistocene of Arabia. In the last ten years, this research has transformed our understanding of Arabia and helped fill one of the largest gaps in our knowledge of SW Asia in the Pleistocene.

The paper shows that hominins occupied Arabia during interglacials when there were rivers and lakes, similar to the Sahara. The demographic history is one of range expansion in periods of higher rainfall, and abandonment when the lakes and rivers dried up under increased aridity. Occupation records are therefore discontinuous, unlike in the northern Levant where populations could have been continuous in areas such as Galilee.

The authors need to add a section in the SI and a brief paragraph in the main text that discuss the wider implications of their Arabian research. If hominins/humans were able to disperse into Arabia at ca. 400, 300, 200, 125-75 ka, they could presumably also disperse out of it into areas further east. My assessment is that this paper considerably strengthens earlier claims for *H. sapiens* in India, SE Asia and S. China before 50 ka; in fact, it considerably strengthens the case that there were multiple dispersals of *H. sapiens* out of Africa (see e.g. Boivin et al. *Quat. Int.* 2013). They might/should discuss the following papers, most of which were published by Nature:

Dispersal in MIS 11 into India: Akhilesh et al. (2018) propose that the Middle Palaeolithic in India began 385 ± 64 ka BP and argue that at Attirampakkam, the gradual disuse of bifaces, the predominance of small tools, the appearance of distinctive and diverse Levallois flake and point strategies, and a blade component all indicate a shift away from the preceding Acheulian large-flake technologies. They also point out that these changes occur within the time range of the long interglacial period of MIS 11. They also suggested that the advent of a Middle Palaeolithic

technology in India at around the same as in Africa may be part of the same process, and there is a possibility that it may have been introduced. Additionally, as the Middle Stone Age in Africa and Arabia is associated with *H. sapiens*, so too it might have been in South Asia with the Middle Palaeolithic.

Akhilesh, K., Pappu, S., Rajapara, H. M., Gunnell, Y., Shukla, A. D. and Singhvi, A. K. (2018) Early Middle Palaeolithic culture in India around 385–172 ka reframes Out of Africa models. *Nature*, 554: 97-101.

H. sapiens in the Thar Desert in MIS 5? Blinkhorn and colleagues (2013, 2015) suggest that groups using a Nubian technology had dispersed as far east as the Thar Desert during MIS 5 and the end of MIS 4 and even into peninsular India

Blinkhorn, J., Achyuthan, H., Petraglia, M. and Ditchfield, P. (2013) Middle Palaeolithic occupation in the Thar Desert during the Upper Pleistocene: the signature of a modern human exit out of Africa? *Quaternary Science Reviews*, 77: 233-238.

Blinkhorn, J., Achyuthan, H., and Ajithprasad, P. (2015) Middle Palaeolithic point technologies in the Thar Desert, India. *Quaternary International*, 382: 237-249.

H. sapiens in India before the Toba super-eruption?: In the Jurreru Valley, cores were not only flaked in the same way before and after the eruption but were technologically similar to the way cores were flaked in sub-Saharan Africa, southeast Asia and Australia, "suggesting modern humans may have entered India before the Toba eruption as part of an early eastward dispersal from Africa" (Clarkson et al. 2012: 165).

Clarkson, C., Jones, S. and Harris, C. (2012) Continuity and change in the lithic industries of the Jurreru Valley, India, before and after the Toba eruption. *Quaternary International*, 258: 165-179.

H. sapiens and microliths in India at 55 ka, and the S. African connection: : the "impossible coincidence"?:

Mellars, P., Gorí, K. C., Carr, M., Soares, P. A. and Richards, M. B. (2013) Genetic and archaeological perspectives on the initial modern human colonization of southern Asia. *Proceedings of the National Academy of Sciences USA*, 110(26): 10699-10704.

The authors might also like to consider claims of *H. sapiens* in Sumatra at 63-73 ka (Lida Ajer) and in S. China pre-80 ka (Fuyan, Zhirendong)

Westaway, K. E., Louys, J., Dur Awe, R., Morwood, M. J., Pricer, G. J. et al. (2017) An early modern human presence in Sumatra 73,000–63,000 years ago. *Nature*, 548: 322–325. doi:10.1038/nature23452.

Liu, W., C. Jin, Y., Zhang, Y., Cai, S., Xing, J., Wu, H., Cheng, R.L., Edwards, W., Pan, D., Qin, Z. An, Trinkaus, E. and Wu, X. (2010a) Human remains from Zhirendong, South China, and modern human emergence in East Asia. *Proceedings of the National Academy of Sciences USA*, 107: 19201-19206.

Wu Liu, Martínón-Torres, M., Cai Yanjun, Xing Song, Tong Haowen et al. (2015) The earliest unequivocally modern humans in southern China. *Nature*, 526: 690-700.

The authors may also wish to make the point that is exceptionally improbable that there was only one dispersal even across Asia at ca. 50-60 ka

Minor comments:

Main text: Fig 1 – a map of last interglacial rainfall would be more useful for Fig. 1b

p. 10, hippos – mention in the main text that they require water 2m deep and abundant riverine vegetation. The authors more or less state this on p.37 of the SI, but general readers need to know this in the main text.

p. 11, end: "seemingly followed by repeated regional depopulation". Add "under increasing aridity"

Author Rebuttals to Initial Comments:

Referee #1 (Remarks to the Author):

The subject matter of this paper is of great and broad interest focussing as it does on trying to bridge the gap between the African Archaeological record where human dispersals are thought to have occurred to Levant and Asian records where people migrated to. To do this the authors have combined a wide range of lake evidence with associated lithic assemblages from the Arabian peninsular. The underlying evidence is fascinating and compelling and of great potential significant and the conclusions appear robust and exciting.

We thank the reviewer for their kind and supportive remarks.

As detailed in comment below I do have some concerns about the chosen 'green' windows.

We have added a new SI section (SI 11) to clarify our findings and interpretations in light of the suggestions and questions from reviewer 1 (first part of SI 11) and reviewer 3 (latter part of SI 11).

We thank the reviewer for their useful comments and address their specific queries below. Our key point is that we find evidence for 'green windows' (i.e. evidence for regional humidity indicated by episodes of lake formation and human occupation in a generally arid region), we date these as best we can, and then we briefly situate these in terms of wider regional knowledge on climate. So we do not feel that we have 'chosen' particular windows as such, rather we have simply reported our results and described their context. Attributing the chronology to particular phases of increased humidity involves both reporting the age estimates themselves, but also interpreting these in their stratigraphic context (i.e. some age estimates are from within lake sediments themselves, others lie above and below the lake sediments providing 'bracketing' ages for increased humidity), and finally situating these results in the contexts of regional climate. In regards to the latter, if, for instance, a known regional humid period is bracketed by long arid periods either side, this supports attribution of a particular lacustrine deposit to that humid period even when the error ranges are large, as it is highly unlikely that lakes formed during periods which were known to be arid. We appreciate the points by the reviewer though, and as outlined below, we have caveated and clarified our arguments.

Abstract: Windows not ranges. Some in huge interglacials other e.g. 55 ka very brief green periods...

The major wet periods in Arabia – as known from multiple archives – do seem to correspond with interglacials. However, there is also evidence for hominin presence in Arabia in glacial periods such as late MIS 5 and early MIS 3. Furthermore, it is clear from the palaeoclimatic record of both Arabia and North Africa that even "interglacial" humid periods do not correspond exactly with the duration of the interglacial in question. For example, the well documented 'African Humid Period' in the Sahara (e.g. deMenocal et al., 2000, QSR) ended ca. 5 ka, but the Holocene interglacial continues to the present day. Consequently we have revised our SI wording to focuss on the role of obliquity modulated precession in strengthening monsoonal systems leading to increased Arabian humidity. In the main text we focus on the approximate age in ka, rather than attribution to particular climatic phases. A

more detailed explanation of the changes made to the manuscript and SI regarding this point is provided below, in response to another of Reviewer 1's comments.

Regarding the specific comment made by Reviewer 1, we acknowledge that the periods of reduced aridity identified in the abstract probably vary in duration and intensity. The value of the KAM-4 sequence, and our reconstruction of a composite record for Arabia, is that no such records exist in this region, but the framework being erected here will underpin future research and will no doubt be refined as new data are acquired. However, the chronology for KAM-4 presented in this paper is not sufficiently precise to allow us to specify time windows. Instead, we have allocated approximate ages to these humid periods to allow the reader of the abstract a clear understanding of the timeframe being considered. The main text of the manuscript, and the SI, provide a much more detailed account of the uncertainties and caveats associated with these age attributions. Consequently, we have retained the original phrasing of this portion of the abstract.

P3 L4/5 Saharo-Arabian and Palearctic biomes implies a north-south migratory dynamic but the paper is more about west-east migrations? Especially as next sentence is arguing for the importance of Asia in Neanderthal-sapiens mixing.

The different biomes have a mostly latitudinal distribution, with the Saharo-Arabian biome covering a strip from the northern part of Africa to the Thar Desert of India (we follow here papers such as Holt et al. (2012; Science) and Stewart et al (2019, Quaternary International), but it is easier to disperse within a biome (i.e. longitudinally) rather than cross into different biomes, with their different climate, fauna etc. So ultimately what we are saying is that amelioration of the central part of the Saharo-Arabian area (i.e. NE Africa and Arabia) allowed populations to move north far enough that they could leave (or enter) Africa, this then allowed them to move east within a similar biome type. This is indicated by factors such as the increasingly African character of the Arabian fossil record. In other words the northwards movement of the ITCZ and the amelioration of the Saharo-Arabian belt encouraged latitudinal dispersal into it, but then rather than continuing north into the Palearctic biome, it is likely that groups followed similar conditions eastward...additionally, understanding these dispersal processes is about a combination of climate, biome and geography. I.e. that the peninsula character of Arabia cuts the Saharo-Arabian arid belt in such a way that only significant northwards movement within Africa allows entry and then eastwards movement.

Intro I felt could be more focussed. The key argument is there are thought to have been dispersals and in and out of Africa but evidence of the pathways between sources and destinations has been hampered but the paucity and fragmentary nature of evidence in two key areas- the Sahara for north-south migrations and The arabian peninsula for east-west migrations.

We take the reviewers point, and have made minor changes to the text (see track changes). We point out, however, that our paper is not just about hominins moving between Africa and Asia and there not being much known in between. We emphasise the evidence for considerable variation within different areas of Southwest Asia, for instance. It was already clear that some areas saw relatively sustained occupations, lasting long enough to give rise to distinct local forms of material culture, such as the Acheulo-Yabrudian in the Levant and the various forms of localized Middle Palaeolithic in southern and eastern Arabia. There are also

issues such as the southwards movement of Neanderthals (how far south?), and debates on palaeoclimate in the area (how far north did tropical, monsoonal, precipitation reach? What role did winter westerly precipitation play in bringing precipitation to Arabia during humid periods?). Our point is that while hominin dispersal out of (and perhaps back into) Africa is an overarching theme in our paper, another is simply understanding the palaeoenvironmental and palaeoanthropological records of Arabia and Southwest Asia more generally on their own terms.

P4 L1 should it be Achuelean culture not just Achuelean?

Either are considered acceptable.

P4 Fig 1 – “A” could be improved but showing PE not just P as this is of more relevance for the existence and persistence of surface water. Would also benefit either for inset showing where in world fig is of and/or some place names to aid readers locate the area easily. Grey areas are sea but could also be made clearer.

We thought about how to improve this figure, and note that reviewer 3 also suggested changes. In the end we decided against trying to factor evaporation into this figure. This is for several reasons. Factors such as cloud cover which would have strongly influenced evaporation are uncertain for the time period under study, and precipitation has been more of a focus for climate model studies in the region, so we prefer to stick with precipitation. Even today, evaporation estimates for areas like Arabia are problematic at a local scale, as so much of it depends on geology. For instance, studies (e.g. Schulz et al 2016, *Hydrological Processes* 30, 771-782) in karstic regions of Saudi Arabia (limestone makes up a large proportion of Arabia) have shown that around half of precipitation rapidly enters underground aquifers. This would obviously be very different in areas with steep volcanic bedrock and limited soil cover. While it is a topic requiring more research, we have suggested that dunefields also acted like sponges and much of the water falling as precipitation soaked into sand and was therefore protected from evaporation. We do agree with the underlying sentiment from the reviewer though, which is why we include the biome map (1A), as this reflects how climate ‘plays out’ in environmental terms (i.e. reflecting factors such as evaporation, topography, etc). We use precipitation as a simple indication for climate change, since this is a relatively reliable model output (Otto-Bliesner et al., 2006), but then acknowledge that how this resulted in changing water availability in the landscape would have been complex. We followed reviewer 3’s suggestion on changing the figure, as discussed below. Finally, we discussed further changes with the professional graphic designer who helped with the figure. We agree with her that the sea colour is good as it is, and that adding place names etc to figures that are already complicated would be messy. The figures have latitudes and longitudes marked on them so they can be easily situated.

P5 17/8 refer interchangeably to sequences and formations. They are not the same thing and could be rephrased to aid readability.

We changed the wording, to talk about the site instead of the sequences.

P5 117 would be helpful if SI3 also was annotated to indicate the carbonate rich marls

SI 3 refers to the section of the supplementary information, not to supplementary figure 3.

P5 122 – I don't quite follow the argument that because the sediments lack coarser material the archaeology is *in situ*. Probably just needs rephrasing to be clearer.

We added a few sentences here to clarify; "...reflecting deposition under low-energy/still-water conditions. Larger clasts (i.e. gravels) are absent, emphasising the lack of higher energy current flow processes feeding the basin during sediment accumulation. Reworking of lithics and fossils from the surrounding landscape into these lake bodies is therefore highly unlikely. Consequently, we argue that the assemblages of lithics and the fossils found in association with these deposits are *in situ*"

P6 16 late Quaternary is being used as a specific time period so should therefore be "the Late Quaternary".

We changed late Quaternary to 'later Middle Pleistocene and Late Pleistocene'. We keep the small l as 'later Middle Pleistocene' is not a formal geological period, but a useful reference to the later part of the Middle Pleistocene, which covers a long time period.

P6 18-24. The errors on the luminescence ages are large (+/-87 thousand years for the oldest) so I find it unclear how attribution to very precise green windows to 400 300 etc can be warranted. This needs explanation.

We agree that the oldest luminescence estimate in particular has a very large uncertainty, being near the effective limits for the technique. For the younger time periods, the uncertainties are much lower and we can be more confident on attribution to particular climate phases.

Our point here is that the green windows are indicated by the presence of lakes themselves. We report the full uncertainty for the chronometric age estimates in the main text and SI, while referring to 'approximately 400 ka' etc as shorthand in the introductory paragraph etc. When we say a lake approximately 400 ka, we are not suggesting that it was exactly 400 ka, but approximately around this time. MIS 11 dates from around 424 to 374 ka, so we think the evidence suggests a most likely attribution to this period, and 400 ka being a reasonable shorthand approximation.

Two points are significant here. Firstly, it should be noted that estimating the actual ages of the lakes requires an interpretation of the chronometric age estimates. For instance, some of the age estimates are within lacustrine sediments themselves, and therefore directly date the lake. Other samples were taken from material underlying lacustrine sediments, and therefore provide maximum age estimates. There is also the point that relative chronology provided by stratigraphic relationships (i.e. partly overlapping sediment bodies) brings further context to the chronometric age estimates. If we take the oldest (ca. 400 ka) estimate, for instance, it is important to note here that the luminescence estimate is from an underlying sand, meaning that it is a maximum estimate for the age of lake formation, which is therefore less likely to date to the older end of the age range. Likewise, it is overlain by the ca. 300-350 ka dated deposits. However, we agree with the reviewer that the uncertainties on the ages for the earlier deposits are large, so we added the caveat that the "age estimates for both the Central and Northeast lakes have large uncertainties".

The second key point here is about the wider regional climatic context, which we have tried to summarise in figure 2. This is significant as it allows us to situate our dates in terms of

what is already known, which allows us to constrain the age of lacustrine deposits, even where the individual chronometric estimates have large associated uncertainties. Climate models consistently suggest that the major humid periods in Arabia reflect the northward expansion of tropical rainfall (e.g. Jennings et al., 2015; Quaternary International). Along with the well dated climate records of southern Arabia (e.g. speleothems, where U-series dating has much lower errors than the dates we report, e.g. Nicholson et al 2020, QSR). Together these models and the existing knowledge on Arabian palaeoclimate allow us to suggest most likely periods within the uncertainties on age estimates. If we take the ‘approximately 400 ka’ estimate, it is clear that the region saw long periods of aridity either side of this time (MIS 11). There is a large gap in speleothem formation in southern Arabia either side of MIS 11 (e.g. Nicholson et al 2020, QSR – and at KAM-4 we are not talking about ephemeral humidity like that known from MIS 3, but a fairly large lake being established, one of many at this time (see also e.g. Rosenberg et al., 2013, QSR, Scerri et al., 2021, Scientific Reports). We can look at records such as East Mediterranean sapropels which relate to the strengthening of monsoonal rainfall to the south, and there is a huge gap in their formation either side of MIS 11. These factors therefore suggest that describing the Central Lake as being approximately 400 ka, while acknowledging the large uncertainties, is reasonable. The same points apply to the ‘approximately 300 ka’ lake at KAM-4, which is probably approximately 300-330 ka based on our age estimates and the regional climatic picture. There is likewise again evidence for aridity either side of MIS 9 in the region, and so we think the age of the lake most likely falls around the mid-point of the luminescence estimate. We emphasise again though that we fully report all uncertainties on age estimates, etc., so the shorthand ‘approximately 400 ka’ is just used in places like the introductory paragraph.

Along these lines explanation is required as to why attribution of Lake events places them in interglacials (e.g MIS 11,9,7) when clearly the current interglacial is hyperarid and not conducive to lake formation. Heavy reliance is apparently made on the wet-dry index which is of North African/Mediterranean origin so not regional. Central and NW Lake could have been ascribed to MIS 10 and 8 respectively or even MIS 12 and 10. Later lakes appear to be associated with cooler MIS stage 6. If these lakes were truly short-lived windows of opportunity then understanding PE through time is important as wet periods when temperatures in the Arabian peninsular were also extreme would not allow lake formation but wet periods during cooling episodes would.

We thank the reviewer for making us think carefully about how to describe our findings and think about their wider context and implications. We discuss this in the new SI 11. We use the term interglacials to refer to ‘true interglacials’ (i.e. MIS 5e is the interglacial within MIS 5...see e.g. <https://agupubs.onlinelibrary.wiley.com/doi/10.1002/2015RG000482>), and have made more explicit our position regarding the causal mechanism for humid periods in Arabia, namely a strengthened monsoon associated with increased boreal summer insolation resulting from obliquity modulated precession. This explanation is consistent with numerous climate models which consistently suggest a southern (i.e. monsoonal) origin for precipitation during Arabian humid phases and with the speleothem record of southern Arabia (e.g. the Nicholson et al QSR paper, that we reference). It is worth noting that the highest precessional peaks, which appear to correspond to the most pronounced Arabian humid periods, occur during interglacials.

We believe this revision addresses all of the anomalies identified by Reviewer 1 above. For example, although the Arabian Peninsula is currently arid, there was a Holocene Humid Period (ca 10 to 6 ka) linked to the early Holocene precessional peak, during which lakes formed in northern Arabia. Similarly, the precessional peak in late MIS 6 (ca. 150 ka) may correlate with the age of the Southwest Lake. However, given that this is only around 10 kyr before the transition to MIS 5 (not very long in terms of the uncertainties on luminescence age estimates), and that full lake formation in the sequence overlies the dating sample, it is possible that this lake dates to the transition to MIS 5 rather than MIS 6. Equally, we note that the Southwest Lake is also the only one at KAM-4 not associated with a lithic assemblage, suggesting that it is may be a more ephemeral humid phase than the others? Lastly, evidence from other sites that humans were in Arabia during MIS 3 is used to support our suggestion that South Lake dates to MIS 3 i.e. that there was a humid period at this time. However, the South Lake sedimentary sequence is very thin, and previous studies have not identified lake formation of this period in the western Nefud (Rosenberg et al., 2013, QSR) suggesting that it was a relatively subdued humid period. This observation is consistent with the relatively weak precessional peaks which occurred during MIS 3.

We acknowledge the reviewer's point that the age estimates for the older lakes have large uncertainties. Consequently, in the main text we only discuss the probable MIS attribution in the context of the dating/stratigraphy section. In the archaeological section of the paper, and for figure 3, we use only approximate ages, i.e. we reduced the focus on climate phases and emphasise the actual age estimates themselves mor.

P10 I22 refrain from stating “perhaps related to the ca. 300-500 ka sediments. Or to stratigraphically underlying deposits” – its speculative and a point is still made that you found hippo remains at the site.

Ok. We were trying to point out the distinct possibility that these remains are older than those from KAM-4, we agree that it is rather speculative given the stratigraphic complexity of the site, so we have removed the sentence “perhaps related to the ca. 300-500 ka sediments. Or to stratigraphically underlying deposits”.

P10 I24 “iconic evidence” phrasing not justified.

We changed to ‘powerful evidence’

P10 Not my area of specialism but the use of component analysis to tease out affinities looks strong and important for the paper – could it be included somehow in Fig 4 rather than in SI

Unfortunately, space prevents this. But we thank the reviewer for their supportive comment.

P12 I1 Unclear how these phased of decreased aridity have been identified or what they are by this point in the paper. Longer-term records of aridity in the peninsular like that of Pruesser et al. or off-shore cores have not been drawn upon. Evidence of the material cultural differences between phases I am guessing is presented in Fig 4 but could be highlighted more.

The phases of decreased aridity are the humid phases that we have identified, which are consistent with the wider paleoclimate picture from Arabia, and the region more generally. The material culture differences are those that we discuss in the previous paragraph and

earlier in the paper, and visually summarised in figure 3 (previously incorrectly labelled as 4). These can be summarized as two phases of Acheulean technology, the youngest with a Levallois component, and then three distinct phases of Middle Palaeolithic technology. To clarify we added the sentence to the last paragraph: “with two phases of Acheulean technology and then three distinct forms of Middle Palaeolithic”

Ex Fig 3 – shows good agreement between U and luminescence dating techniques and good increase in age with depth for most sites. Its unclear why some ages are boxed with two ages in which do not agree in some instances particularly jsm-1. Some symbology to get across which age is from what technique would be helpful.

The boxed ages are where there are multiple grain populations, with each quoted age being a major population identified through the Finite Mixture Model, whereas in other cases the ages are modelled with the Central Age Model. We included symbols distinguishing quartz and feldspar luminescence ages in figure 2 in the main text.

Its unclear for figures or text which ages are based on feldspar/IRSL and which temperature and which quartz/OSL. Also in SI no indication was made if a residual dose was subtracted from what is quite a hard to bleach pIRSL290 or pIRSL 225 signals.

As noted above, the figures have been clarified to indicate which mineral was used to produce each date. Residual doses were not subtracted from the pIRSL equivalent doses prior to age calculation since a modern Aeolian sand from KAM-4 yielded negligible residual signals. This point has been clarified by adding the following paragraph to section 5.4 of the Supplementary Information:

“Although they exhibit limited fading, the pIRIR₂₂₅ and pIRIR₂₉₀ signals are reduced to zero much more slowly than quartz OSL or lower temperature (e.g. IR₅₀) K-feldspar emissions. Consequently, it is possible that these signals may yield age overestimates due to the presence of a “residual” or “unbleached” signal in K-feldspar at the point of burial. To test this possibility, K-feldspar from a modern dune sample (PD19) taken from KAM-4 were measured using the pIRIR₂₂₅ and pIRIR₂₉₀ measurement procedures detailed in Supplementary Table 4. This analysis yielded residual doses of 1 ± 0.2 Gy for pIRIR₂₂₅ and 2.5 ± 0.6 Gy for pIRIR₂₉₀ (1σ , $n=6$ in both cases). Since PD19 is believed to be analogous to the material incorporated into ancient samples from KAM-4 and elsewhere in the Nefud Desert at the time of burial, pIRIR ages in this study were not corrected for the presence of an unbleached residual dose.”

Ex fig 4-5 beautifully taken photos!

Thanks.

Referee #2 (Remarks to the Author):

This is a remarkably important paper that addresses an equally important question: What was

the role of SW Asia in the dispersal of hominins out of Africa. As the authors note, perhaps a bit too emphatically, defensively, the traditional narrative has been that hominins remained in Africa *sensu stricto* until ~70 ka and burst out, traversed across Arabia and the Levant and populated Europe, SE Asia, and the rest of the world.

We thank the reviewer for their supportive comments. We have added in SI 11 more on the implications of our findings, particularly the point that we contribute to a growing pool of evidence that there were multiple phases of population dispersal over a broad time period.

New research by this group and others has refined that story to now tell a more inclusive story that hominins had earlier forays into Arabia extending at least back to MIS Stage 5 and probably earlier. The SW Asia and European fossil, lithic, and genomic data still hold a firm hold on the earliest old world dispersal around 70 ka, but it's clear hominins were in Arabia way before this during co-called Green phases, when orbital forcing of the monsoon favored wetter, vegetated landscapes suitable for habitation. This paper is important as it extends this Arabian habitation story back to at least 400 ka, much earlier than previously appreciated. It's an important result and extremely well documented in terms of the comprehensive geology, dating, archeology, and context of the new results. It's an important addition to the large narrative and worthy of publication in my view.

Thanks.

Aside from the very thorough field documentation (in the 100+ page Supp Mat section), the entire story rests upon thermoluminescence dating of sands containing the artifacts. This method has many uncertainties, and the method is less useful for >300 ka dating, hence the very large 20% uncertainty of the 400 Ka basal occupation site. That said, they fortunately have one tooth where an independent U-series date reveals an age consistent with the thermo dates, and this lends considerable support to the overall study (they not doubt rejoiced). I would like to have seen/see more dates on teeth if they were conducted. Moreover, the lake phases are roughly consistent with known wet phases from the eastern Med and regional speleothem data.

As discussed in relation to reviewer 1, we have added a caveat to the oldest (ca 400 age) luminescence estimate. While the uncertainties are large, we think it important to point out regional evidence for tens of millennia either side of MIS 11, strengthening our confidence that the age of Central Lake is indeed around 410 ka (the central point of the OSL range). For KAM-4 more broadly, we also emphasise the importance of relative chronology, with age estimates being in correct stratigraphic order in partially overlapping sediment bodies. The U-series estimate does indeed add considerably to the chronological narrative. Unfortunately, fossils from the site are poorly preserved and upon evaluation, few seem suitable for U-series and ESR dating. Our previous works in the area have shown the great difficulty to identify samples that may be suitable for direct dating (e.g. Stimpson et al., 2016 QSR; Stewart et al., 2020 QR). So while future work to recover better preserved material for further dating may be conducted, this is not currently possible. We agree with the reviewer that the humid periods we identify correlate well with other regional records.

Altogether, this is a very tight story and I have little to critique on the science.

I do think their supposition that the 'northern (Levant" vs. southern" (Arabia) route differentiation implies some fundamentally different mode of human migration in response to

environmental change is misguided. During these wet phases the hyperacid desert regions were vegetated with large permanent lakes (resources) and thus were corridors for exchange as was the Levant. So, the point to be made with this paper is rather that the Arabian peninsula was an equivalent migration corridor during wet phases - something we've know for decades (McClure, 1976). The current paper puts people into this narrative, with water, dates, multiple times, and very early on. (Note the authors can include early Holocene (6-9 ka) and late Pleistocene (25-30 ka) Lake occurrences on their Fig. 2)

Our point on the difference between northern and southern routes is a relatively minor part of our argument, but never the less one that we do think significant. Given there the Red Sea has been present for millions of years, the southern route would indicate an ability for viable sized human groups to have crossed at least several kilometres of sea (with strong currents, sharks etc). This would suggest an ability to engage with coastal and marine environments, evidence for which is currently lacking in the region in such early time periods. It has also been claimed that if humans followed the southern route, then the role of environmental change in these dispersals may have been limited as much of the arid zone of northeast Africa and northern Arabia could be bypassed (e.g. <https://www.biorxiv.org/content/10.1101/2020.01.12.901694v1.full>). By contrast, dispersals by the northern route are more suggestive of extensive environmental amelioration of the arid belt....Additionally, if the southern route had been followed, we would expect there to be close material culture similarities between East Africa and southern Arabia. As we discuss, this does not appear to be the case, with southern Arabia largely demonstrating seemingly autochthonous characteristics.

...Conversely, however, it should be pointed out that these humid phases are also humid in only a relative sense. Conditions would have been highly seasonal, and precipitation variable. Climate models, the continuation of arid adapted fauna, and the differences in fauna between either side of the arid belt (e.g. contrast the faunal records of East Africa and the Levant) all suggest that even during humid periods such as those of MIS 5, arid areas remained in the region (e.g. Scerri et al., 2014 QSR). As a result, a key role is play by occasional fluvial networks across generally arid regions (e.g. Breeze et al. 2016 QSR). Our point here being that the situation is not one of total and complete amelioration of the region, but rather more of a patchwork. The lakes in our study area reflect feeding by groundwater, and reflect the complex relationship between precipitation, evaporation, and water soaking into the sands and underlying bedrock aquifers. The combination of this environmental evidence with our new archaeological findings suggests dispersal into Arabia by a northern route – also being followed by some species of animals.

Figure 2 includes the Holocene Humid Period. The idea that there was a humid period ca 25-30 ka is now generally thought to reflect problematic old radiocarbon dates, and where luminescence dating has been applied, ages have actually been shown to be much older (see for example Rosenbeg et al (2011, Geology).

Bravo !

Many thanks for your kind remarks.

Referee #3 (Remarks to the Author):

I have no criticism over the contents of the paper, which collates a large amount of

excavation and survey. This is an excellent paper by a long-established high-quality team that has already published several papers on the palaeolithic and Pleistocene of Arabia. In the last ten years, this research has transformed our understanding of Arabia and helped fill one of the largest gaps in our knowledge of SW Asia in the Pleistocene.

We thank the reviewer for their support.

The paper shows that hominins occupied Arabia during interglacials when there were rivers and lakes, similar to the Sahara. The demographic history is one of range expansion in periods of higher rainfall, and abandonment when the lakes and rivers dried up under increased aridity. Occupation records are therefore discontinuous, unlike in the northern Levant where populations could have been continuous in areas such as Galilee.

The authors need to add a section in the SI and a brief paragraph in the main text that discuss the wider implications of their Arabian research. If hominins/humans were able to disperse into Arabia at ca. 400, 300, 200, 125-75 ka, they could presumably also disperse out of it into areas further east. My assessment is that this paper considerably strengthens earlier claims for *H. sapiens* in India, SE Asia and S. China before 50 ka; in fact, it considerably strengthens the case that there were multiple dispersals of *H. sapiens* out of Africa (see e.g. Boivin et al. *Quat. Int.* 2013).

We can't add much to the main paper as we are already at the limit for space/words. We are also somewhat cautious about how far our results can be extrapolated eastwards, given current lack of knowledge on palaeoclimate and early prehistory in areas like Iran. Authors of our paper have long argued that there were multiple dispersals of our species into Asia, including at earlier time periods than often thought (e.g. Petraglia et al., 2007, *Science*; Groucutt, et al., 2015, *Evolutionary Anthropology*.) Very diverse opinions have been expressed on the recent findings from across Asia; some, for instance, have argued that even if there were extensive dispersals as far east as China and Sahul, these 'failed' and did not contribute to more recent populations in these areas (or only contributed at very minor levels). Our paper on Southwest Asia is not the place to try and resolve these debates. We hope that our paper contributes to a growing pool of evidence for multiple dispersals into Asia, as the reviewer points out. Finally, we emphasise that not all of the material from Arabia indicates dispersals from Africa, it is quite possible for instance that in MIS 3, Neanderthals dispersed into the area from the north. So while multiple dispersals of *Homo sapiens* out of Africa are an important part of our findings, the diversity of material culture suggests multidirectional dispersals, and the seemingly repeated regional 'failure' of population expansions also suggests how complicated the situation is just in southwest Asia, let alone more broadly across the vast area of Asia. Most geneticists agree that all humans outside Africa result from a single expansion ca. 60-50 ka. How to reconcile these data is currently unclear.

We added a new SI section (SI 11) discussing the implications of our findings, and we thank the reviewer for this suggestion, which we think is useful. We added most of the papers they mention below, as well as others.

They might/should discuss the following papers, most of which were published by Nature:

Dispersal in MIS 11 into India: Akhilesh et al. (2018) propose that the Middle Palaeolithic in

India began 385 ± 64 ka BP and argue that at Attirampakkam, the gradual disuse of bifaces, the predominance of small tools, the appearance of distinctive and diverse Levallois flake and point strategies, and a blade component all indicate a shift away from the preceding Acheulian large-flake technologies. They also point out that these changes occur within the time range of the long interglacial period of MIS 11. They also suggested that the advent of a Middle Palaeolithic technology in India at around the same as in Africa may be part of the same process, and there is a possibility that it may have been introduced. Additionally, as the Middle Stone Age in Africa and Arabia is associated with *H. sapiens*, so too it might have been in South Asia with the Middle Palaeolithic.

Akhilesh, K., Pappu, S., Rajapara, H. M., Gunnell, Y., Shukla, A. D. and Singhvi, A. K. (2018) Early Middle Palaeolithic culture in India around 385–172 ka reframes Out of Africa models. *Nature*, 554: 97-101.

H. sapiens in the Thar Desert in MIS 5? Blinkhorn and colleagues (2013, 2015) suggest that groups using a Nubian technology had dispersed as far east as the Thar Desert during MIS 5 and the end of MIS 4 and even into peninsular India

Blinkhorn, J., Achyuthan, H., Petraglia, M. and Ditchfield, P. (2013) Middle Palaeolithic occupation in the Thar Desert during the Upper Pleistocene: the signature of a modern human exit out of Africa? *Quaternary Science Reviews*, 77: 233-238.

Blinkhorn, J., Achyuthan, H., and Ajithprasad, P. (2015) Middle Palaeolithic point technologies in the Thar Desert, India. *Quaternary International*, 382: 237-249.

H. sapiens in India before the Toba super-eruption?: In the Jurreru Valley, cores were not only flaked in the same way before and after the eruption but were technologically similar to the way cores were flaked in sub-Saharan Africa, southeast Asia and Australia, “suggesting modern humans may have entered India before the Toba eruption as part of an early eastward dispersal from Africa “ (Clarkson et al. 2012: 165).

Clarkson, C., Jones, S. and Harris, C. (2012) Continuity and change in the lithic industries of the Jurreru Valley, India, before and after the Toba eruption. *Quaternary International*, 258: 165-179.

H. sapiens and microliths in India at 55 ka, and the S. African connection: : the “impossible coincidence”?:

Mellars, P., Gorí, K. C., Carr, M., Soares, P. A. and Richards, M. B. (2013) Genetic and archaeological perspectives on the initial modern human colonization of southern Asia. *Proceedings of the National Academy of Sciences USA*, 110(26): 10699-10704.

The authors might also like to consider claims of *H. sapiens* in Sumatra at 63-73 ka (Lida Ajer) and in S. China pre-80 ka (Fuyan, Zhirendong)

Westaway, K. E., Louys, J., Dur Awe, R., Morwood, M. J., Princes, G. J. et al. (2017) An early modern human presence in Sumatra 73,000–63,000 years ago. *Nature*, 548: 322–325. doi:10.1038/nature23452.

Liu, W., C. Jin, Y., Zhang, Y., Cai, S., Xing, J., Wu, H., Cheng, R.L., Edwards, W., Pan, D., Qin, Z. An, Trinkaus, E. and Wu, X. (2010a) Human remains from Zhirendong, South China, and modern human emergence in East Asia. *Proceedings of the National Academy of Sciences USA*, 107: 19201-19206.

Wu Liu, Martínón-Torres, M., Cai Yanjun, Xing Song, Tong Haowen et al. (2015) The earliest unequivocally modern humans in southern China. *Nature*, 526: 690-700.

The authors may also wish to make the point that is exceptionally improbable that there was only one dispersal even across Asia at ca. 50-60 ka

We thank the reviewer for this point. We added in SI the point that our findings support the presence of multiple 'early' phases of dispersal. What became of these pulses is likely to remain very much debated for a long time yet, but we think our evidence for multiple pulses of dispersal is an important element of discussions.

Minor comments:

Main text: Fig 1 – a map of last interglacial rainfall would be more useful for Fig. 1b

Thanks for the idea, we have updated the figure accordingly.

p. 10, hippos – mention in the main text that they require water 2m deep and abundant riverine vegetation. The authors more or less state this on p.37 of the SI, but general readers need to know this in the main text.

This is a good point. We agree with hippos provide a powerful insight into the environment. We added “which require permanent water several metres deep”.

p. 11, end: “seemingly followed by repeated regional depopulation”. Add “under increasing aridity”

Done.

Reviewer Reports on the First Revision:

Referees' comments:

Referee #1 (Remarks to the Author):

In my opinion the authors have carefully looked at all comments I made in my review and taken them on board. I am completely happy with how they have dealt with the comments and as a result see the paper as complete and ready for publication. It looks to me to be a very significant contribution which will be cited by many and encourage future research to revise and add detail to what these authors have accomplished.

Referee #2 (Remarks to the Author):

I think the authors for their attention to the various ideas and comments raised by reviewers. With regard to my comments on the manuscript, I feel the authors have adequately addressed my concerns. I continue to support the publication of this study as I feel it will provide an important contribution to the larger question of the timing and pathways of human dispersal out of Africa. I feel the authors have sufficiently nuanced the northern versus southern route discussion.

I was also heartened to find the other reviews were as supportive of the publication as I was. Indeed this is a rare outcome for any journal, but especially this one. This is testament to the incredible effort and professionalism of the scientific team. It shows what happens when one aligns adequate funding with great opportunity and exceptional talent.

many thanks for the opportunity to review on this and I look forward to hopefully seeing this in

print soon.

Referee #3 (Remarks to the Author):

I am satisfied by their response to the reviewers, and they have dealt with all queries and comments in an appropriate manner. I am glad that the authors have accepted my recommendation that they should note the wider implications of their paper, and the addition of SI section 11 is very useful.

This is an important paper, and I support its publication. It succeeds in its two main aims: 1) showing the differences between the Levantine and Arabian records – the former, Palearctic, largely continuous occupation, with mainly cave residential sites, and the latter, Afro-Saharan, discontinuous occupation, with mainly open-air workshop sites 2) clearly and emphatically making the case for multiple dispersals out of Africa, as argued by many of us over the last ten years. This should help dispel any adherence to the single-event model for the dispersal of our species across Eurasia.

Main text, Fig. 1 – I note they have now amended this to show last interglacial precipitation.

There are a few minor points in the SI that can be cleared up in proof-reading:

SI, line 5: should be “has been interpreted”

Line 22: “the main admixture between Homo sapiens and Neanderthals, which gives all humans today about 2% Neanderthal DNA⁴⁵”; should this be “non-African humans”?

SI, p.4, line 21: “The oldest claims for human occupation currently comes” – change to “come”

SI, p.11, line 17: “The stratigraphy and dating of the new trenches follows” – change to “follow”

SI, p.13, line 21: “The recombination of the electrons result in the emission” – change to “results”

SI, p.17, line 27: “Cosmic dose rate were calculated” – change to “cosmic dose rates”

SI, p.18, line 7: space needed with “supplementaryfigures”

SI, p.41, line 18: “With the caveat the diverse regional behavioural”; change to “With the caveat that the diverse regional behavioural”

Author Rebuttals to First Revision:

Response to reviews

We thank the reviewers for their kind comments.

We have correct the typos and grammatical mistakes pointed out by review 3.